# Learning the Transformer Kernel

**Sankalan Pal Chowdhury**                                        spalchowd@inf.ethz.ch
**Department of Computer Science**
**ETH Zürich**

**Adamos Solomou**                                        solomou.adamos@gmail.com
**Department of Computer Science**
**ETH Zürich**

**Avinava Dubey**                                        avinavadubey@google.com
**Google Research**
**Mountain View, CA**

**Mrinmaya Sachan**                                        mrinmaya.sachan@inf.ethz.ch
**Department of Computer Science**
**ETH Zürich**

**Reviewed on OpenReview:** `https://openreview.net/forum?id=tLIBAEYjcv`

## Abstract

In this work we introduce KL-TRANSFORMER, a generic, scalable, data driven framework for learning the kernel function in Transformers. Our framework approximates the Transformer kernel as a dot product between spectral feature maps and learns the kernel by learning the spectral distribution. This not only helps in learning a generic kernel end-to-end, but also reduces the time and space complexity of Transformers from quadratic to linear. We show that KL-TRANSFORMERs achieve performance comparable to existing efficient Transformer architectures, both in terms of accuracy and computational efficiency. Our study also demonstrates that the choice of the kernel has a substantial impact on performance, and kernel learning variants are competitive alternatives to fixed kernel Transformers, both in long as well as short sequence tasks. [1]

## 1 Introduction

Transformer models (Vaswani et al., 2017) have demonstrated impressive results on a variety of tasks dealing with language understanding (Devlin et al., 2019; Radford et al., 2018; Raffel et al., 2020; Brown et al., 2020), image processing (Parmar et al., 2018; Carion et al., 2020; Lu et al., 2019), as well as biomedical informatics (Rives et al., 2020; Ingraham et al., 2019; Madani et al., 2020). Albeit powerful, due to the global receptive field of self-attention, the time and memory complexity of Softmax Transformer models scale quadratically with respect to the sequence length. As a result, the application of Transformers to domains with long contexts is rather limited. This limitation has spawned several efficient Transformer designs (Liu et al., 2018; Parmar et al., 2018; Child et al., 2019; Zaheer et al., 2020; Beltagy et al., 2020; Roy et al., 2020; Tay et al., 2020a; Kitaev et al., 2020). Kernelization offers one such design. The use of kernel feature maps allows to reformulate the computation of attention in a way that avoids the explicit computation of the full attention matrix which is the key bottleneck for Softmax Transformer. This also opens up new directions for more generic attention mechanisms.

Tsai et al. (2019) first proposed a kernel-based formulation of the attention mechanism. However, the time and memory complexity of their approach remains quadratic with respect to the sequence length. To address this limitation, Katharopoulos et al. (2020) expressed self-attention as the inner product of kernel feature maps and made use of the

---

[1]Our code and models are available at `https://github.com/cs1160701/OnLearningTheKernel`

associative property to reduce the complexity from quadratic to linear. For their experiments, they used the arbitrarily chosen LinearElu feature map $f(x) = \max(x+1, e^x)$. Performer (Choromanski et al., 2021b) replaces this with feature maps that can directly approximate the softmax kernel, thereby allowing the use of pre-trained Softmax Transformer weights in a linear time model. Concurrently with them, Peng et al. (2021) proposed a linear space and time method that added causal and recency based features to random Fourier methods. More recently, Schlag et al. (2021b) showed the formal equivalence of linearized self-attention mechanisms and fast weight programmers. While the aforementioned approaches provide a significant reduction in computational and memory requirements, this often comes at the cost of performance, as can be seen from Fig. 1. In this work, we posit that this is partly due to the fact that the similarity functions/kernels, including scaled-dot-product, were hand picked and not learnt from data. Thus, we explore whether kernel learning can help to bridge this gap in performance while retaining the scalability of efficient Transformers.

Although, to the best of our knowledge, kernel learning has never been explored within the framework of Transformers, kernel learning methods have been an ubiquitous tool in machine learning. The most notable among them is Random Kitchen Sinks (RKS; Rahimi & Recht, 2007), a data-independent framework for approximating shift-invariant kernels using an explicit feature map. In RKS, the kernel is approximated by $\kappa(x, y) \approx \langle \phi(x), \phi(y) \rangle$, where the explicit feature map $\phi : \mathbb{R}^d \to \mathbb{R}^s$ is obtained by sampling from a spectral distribution defined by the inverse Fourier transform of the kernel function $\kappa$. Wilson & Adams (2013) modeled the spectral density as a mixture of Gaussians, A la Carte (Yang et al., 2015) proposed an optimization based framework for learning the spectral distribution, BaNK (Oliva et al., 2016) modeled the spectral distribution using an infinite mixture of Gaussians, while Fang et al. (2020) implicitly modeled it using deep generative models. We build on these advances and incorporate them into the Transformer framework.

**Contributions:** In this work, we propose KL-TRANSFORMER, a scalable data driven framework for learning the kernel of Transformers and investigate whether a fully learnable kernel can help to improve the performance of linear, fixed kernel Transformers. Thus, we introduce Transformers with learnable similarity functions, which happen to retain the linear complexity in terms of the sequence length. We motivate our learning method using RKS and learn the kernel by learning the corresponding spectral distribution. In §2.1 we first propose to learn a generic Transformer kernel by explicitly approximating the spectral distribution using a Mixture of Gaussians (GMM) and propose modifications to scale it further. In an attempt to further explore the trade off between computational complexity and accuracy we also propose to model the spectral frequency distribution of Transformer kernels implicitly by using deep generative models (Goodfellow et al., 2014). Finally, we also propose a novel method to learn the spectral distribution of positive random feature (PRF) maps, which provides a better approximation of the softmax kernel (Choromanski et al., 2021b).

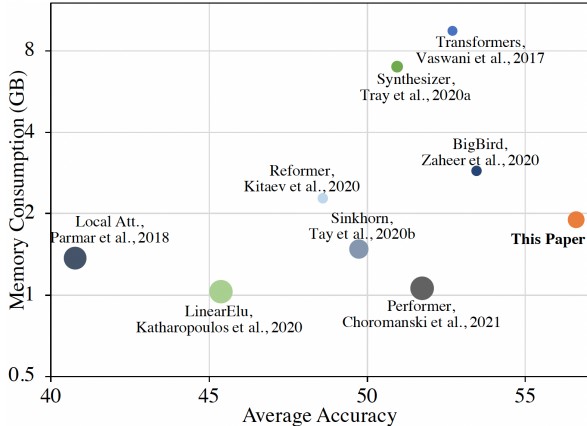

Figure 1: Peak memory (y-axis), average performance (x-axis) and speed (denoted by area of circle) for various efficient Transformer models (i.e. bigger circles in the bottom right corner are better) across three long sequence tasks ($> 1024$ tokens) introduced in *LRA* (Tay et al., 2021b). All values except for "This Paper" are taken from Tay et al. (2021b).

We analyse the expressivity and precision of our proposed models (§2.2) and show that the proposed GMM with positional encodings is Turing-complete (Pérez et al., 2019) with controllable variance. We experimentally evaluate our models on *LRA* (tasks with long context), *GLUE* (tasks with short context) and a synthetic dataset with controllable sparsity, and analyze the performance of our models (§3, §2.2). In our experiments, we find that learnt kernels improve performance in long-context tasks, while staying competitive to the Softmax Transformer of the same size in short-context tasks.

We also find that our models learn parameters that reduce the variance in their predictions, and can handle sparsity quite well. We also benchmark the computational efficiency of KL-TRANSFORMERs and find that each of our proposed KL-TRANSFORMERs scales linearly with respect to the sequence length. We conclude with a short comparison between Random Kitchen Sinks (RKS, Rahimi & Recht (2007)) and Positive Random Features Choromanski et al. (2021b) in terms of their performance and provide recommendations on which approach should be chosen under which circumstances.

## 2 Kernel Learning in Transformers

We begin with the generalized formulation of self-attention proposed by Tsai et al. (2019). Given a non-negative kernel function $\kappa(\cdot, \cdot) : \mathbb{R}^{d_k} \times \mathbb{R}^{d_k} \to \mathbb{R}_+$, the output of the generalized self-attention mechanism at index $i$ operating on an input sequence $X = (x_1, ..., x_L) \in \mathbb{R}^{L \times d}$ is defined as

$$\text{ATTN}(X)_i = \sum_{j=1}^{L} \frac{\kappa(q_i, k_j)}{\sum_{j'=1}^{L} \kappa(q_i, k_{j'})} v_j. \tag{1}$$

where $k_i = x_i W^K, q_i = x_i W^Q, v_i = x_i W^V$ are linear transformations of the input sequence into keys, queries and values of dimension $d_q = d_k$ and $d_v$ respectively and $W^K \in \mathbb{R}^{d \times d_k}, W^Q \in \mathbb{R}^{d \times d_q}, W^V \in \mathbb{R}^{d \times d_v}$. While the formulation in Eq. (1) is more generic and defines a larger space of attention functions, it suffers from a quadratic time and memory complexity. To reduce the quadratic time and memory, we briefly review the method of random Fourier features for the approximation of kernels (Rahimi & Recht, 2007). The details of the method will help motivate and explain our models.

**Random Fourier Features for Kernels:** At the heart of this method lies the theorem of Bochner (Rudin, 1990) which states that a continuous shift invariant kernel $\kappa(q, k) = \tilde{\kappa}(q - k)$ over arbitrary variables $q$ and $k$ is a positive definite function if and only if $\tilde{\kappa}(\delta)$ is the Fourier transform of a non-negative measure $\rho(\omega)$. Moreover, if $\tilde{\kappa}(0) = 1$, then Bochner's theorem guarantees that $\rho(\omega)$ is a normalized density function, i.e.

$$\tilde{\kappa}(q - k) = \int_{\mathbb{R}^d} \rho(\omega) \exp\left(i\omega^T(q - k)\right) d\omega = \mathbb{E}_{\omega \sim \rho}\left[ \exp(i\omega^T q) \exp(i\omega^T k)^* \right]. \tag{2}$$

Rahimi & Recht (2007) proposed to sample from the spectral density $\rho(\omega)$ for a Monte Carlo approximation to the integral in Eq. (2). Specifically, for real valued kernels, they define $\kappa(q, k) \approx \phi(q)^T \phi(k)$, where $\omega_i \sim \rho(\omega)$ and

$$\phi(x) := RKS(x, \Omega = (\omega_1, \ldots, \omega_M)) := \frac{1}{\sqrt{M}} [\cos(\omega_1^T x), \ldots, \cos(\omega_M^T x), \sin(\omega_1^T x), \ldots, \sin(\omega_M^T x)] \tag{3}$$

To learn a kernel, we can either learn a parametric function $\kappa(\cdot, \cdot)$ or learn the corresponding parameterized feature map $\phi(\cdot)$ directly, which corresponds to learning the spectral density $\rho(\omega)$ (Wilson & Adams, 2013; Yang et al., 2015; Oliva et al., 2016). In this paper, we focus on the latter because this helps us in keeping the computational complexity linear in the sequence length $L$. This can be achieved by rewriting Eq. (1) as $\text{ATTN}(X)_i = \frac{\phi(q_i)^T (\sum_{j=1}^{L} \phi(k_j) v_j^T)}{\phi(q_i)^T \sum_{j'=1}^{L} \phi(k_{j'})}$. To the best of our knowledge this is the first attempt to learn the kernel of the generalized self-attention mechanism (Eq. 1).

### 2.1 Learning Kernels in Spectral Domain

**GMM-RKS:** Our objective is to enable learning of any translation invariant kernel. This is realizable if we can learn the spectral distribution. Gaussian Mixture Models (GMMs) are known universal approximators of densities and hence may approximate any spectral distribution. GMMs have been shown to be useful for kernel learning for regression and classification tasks (Wilson & Adams, 2013; Oliva et al., 2016). Thus, to learn the kernel of the generalized self-attention mechanism (Eq. 1), we model the spectral distribution of the kernel as a parameterized GMM, i.e.

$$\rho(\omega) = \sum_{c=1}^{C} \pi_c \mathcal{N}(\mu_c, \Sigma_c) \Leftrightarrow \kappa(q, k) = \sum_{c=1}^{C} \pi_c e^{(i\mu_c^T(q-k) - \frac{1}{2}(q-k)^T \Sigma_c(q-k))} \tag{4}$$

Here $\{\mu_c \in \mathbb{R}^d, \Sigma_c \in \mathbb{R}^{d^2}\}_{c=1}^{C}$ are the learnable parameters of the feature map and $C$ is the number of components in the Gaussian mixture. It can be shown using Plancherel's Theorem that $\rho(\omega)$ can approximate any shift invariant kernel (Silverman, 1986). Since we are working with only real valued kernels, the corresponding kernel reduces to $\kappa(q, k) = \sum_{c=1}^{C} \pi_c e^{(-\frac{1}{2}(q-k)^T \Sigma_c(q-k))} \cos\left(\mu_c^T(q-k)\right)$.

To speedup learning, we assume that $\pi_c = \frac{1}{C}$ and parameterize the feature map with spectral frequency, $\Omega = (\omega_{c,1}, \ldots, \omega_{C,M})$ as:

$$\phi_{\text{GMM-RKS}}(x) := RKS(x, \Omega), \quad \omega_{c,m} = \Sigma_c n_m + \mu_c, \quad n_m \sim \mathcal{N}(\mathbf{0}, \mathbf{I}). \tag{5}$$

This allows us to sample $n_m \sim \mathcal{N}(\mathbf{0}, \mathbf{I})$ and learn the parameters of the feature map, $(\{\mu_c \in \mathbb{R}^{d_q}, \Sigma_c \in \mathbb{R}^{d_q^2}\}_{c=1}^{C})$ end-to-end along with the other parameters of the Transformer.

**FASTFOOD-RKS:** GMM-RKS removes the quadratic dependency on context length, but we still need to calculate $\Omega^T \mathbf{Q}$ and $\Omega^T \mathbf{K}$ (where $\Omega = [\omega_1, \omega_2, \dots, \omega_M]$) which takes $\mathcal{O}(M d_q L)$ time and $\mathcal{O}(M d_q + d_q L + M L)$ space, which can be too much if $M$ is large. For further gains in scalability, we approximate the spectral frequency matrix $\Omega$, using the product of Hadamard matrices (FastFood; Le et al., 2013), such that the computation can be done in time log-linear in $M$, i.e.:

$$\phi_{\text{FASTFOOD}-\text{RKS}}(x) := RKS(x, V), \quad \text{where } V = \frac{1}{\sigma \sqrt{d_q}} SHG\Pi HB. \tag{6}$$

Here, $\Pi \in \{0, 1\}^{d_q \times d_q}$ is a permutation matrix, $H$ is the Walsh-Hadamard matrix, $B$ is a diagonal random matrix with $\{\pm 1\}$ entries, $G$ is a diagonal matrix with Gaussian entries drawn from $\mathcal{N}(0, 1)$ and finally $S$ is a random diagonal scaling matrix that makes the row lengths non-uniform. The entire multiplication can be carried out in logarithmic time, and the space requirement is reduced by storing diagonal matrices instead of full matrices. For $M > d_q$ we use multiple blocks, and the only restriction is that we need $M | d_q$. In order to make this learnable, Yang et al. (2015) proposed making $S$ and optionally $G$ and $B$ learnable. For the main paper, we keep all three learnable (the case where only $S$ is learnable is discussed in Appendix D).

**GENERATIVE-RKS:** If we increase the number of components ($C$) in GMM-RKS, the computation and space complexity increases dramatically. Instead, to learn a more generic kernel, without blowing up the computational complexity, we use deep generative models (DGMs). DGMs have achieved impressive results in density estimation (Goodfellow et al., 2014; Kingma & Welling, 2014; Richardson & Weiss, 2018; Ruthotto & Haber, 2021) and end-to-end kernel learning in the spectral domain for classification (Fang et al., 2020).

In GENERATIVE-RKS we use a DGM to replace the Gaussian probability distribution from GMM-RKS with an arbitrary probability distribution. This DGM acts as a generator similar to the ones used by variational auto-encoders used in computer vision (Kingma & Welling, 2014). In particular, to make the sampling process differentiable, we use the reparameterization trick, where a learnable neural network (called the generator, and denoted by g in Figure 2) transforms samples from a simple noise distribution ($\rho_0$ in Fig 2) into samples from the desired distribution.

$$\omega_m = g(n_m), \quad n_m \sim \rho_o(\cdot) \tag{7}$$

The generator network is trained end to end with the whole model, allowing it to choose the best possible distribution for the given data. These samples ($\Omega = [\omega_1, \omega_2, \dots, \omega_M]$ in Fig 2) are then used in Random Kitchen Sinks as follows:

$$\phi_{\text{GENERATIVE}-\text{RKS}}(x) := RKS(x, \Omega), \quad \omega_m \sim g(\rho_o) \tag{8}$$

We experimented with various configurations and eventually and chose to learn a generator network which consisted of 4 fully connected layers with batch normalisation and LeakyReLU activation, followed by a single fully connected layer with $\tanh$ activation. While this methodology allows us to generalise the gaussian distribution to a much larger class of distributions, it also causes a blowup in the number of parameters e.g. a $4 + 1$ layer constant width generator, as used by us would require $5d^2 + 5d$ parameters as opposed to the $d^2 + d$ parameters in GMM-RKS. To counter this and to improve generalization, we share the same generator network across all heads in a layer, which means that the different heads only differ in the Query/Key/Value projection matrix.

**Positive Random Features (PRF):** Until now, we focused on feature maps defined using RKS. While our formulation is very general, recently it was shown that positive random features provide a better approximation to both Gaussian and Softmax kernels (see Lemma 2 in Choromanski et al. 2021b). In particular they showed that $\kappa(q, k) = \exp(q^T k) = \mathbb{E}_{\omega \sim \mathcal{N}(0, I)}[\exp(\omega^T q - \frac{\|q\|^2}{2}) \exp(\omega^T k - \frac{\|k\|^2}{2})]$ and demonstrated that Monte Carlo approximation to this expectation leads to a low variance estimate of the softmax kernel. Moreover, the presence of only positive values within the randomized feature map ensures that kernel estimates remain strictly non-negative. To incorporate this prior knowledge, we propose a novel kernel learning framework in which we learn the spectral density while using the feature map corresponding to the above expectation. For instance, when we model the spectral distribution of $\Omega = (\omega_1, \dots, \omega_M)$ using GMM ($\psi = \sum_{c=1}^C \pi_c \mathcal{N}(\mu_c, \Sigma_c)$) we have that:

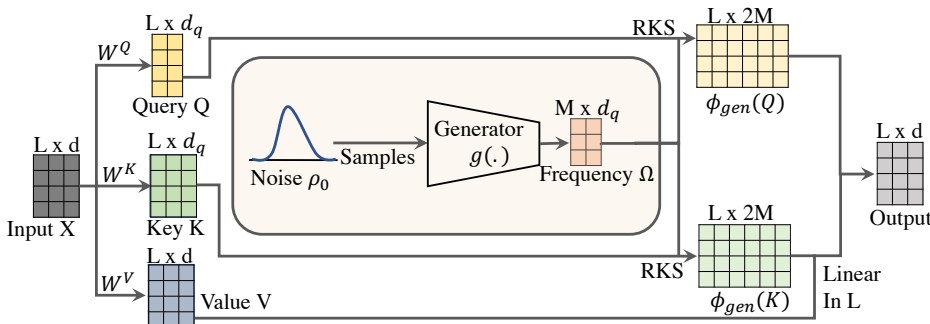

Figure 2: Generalized self-attention with deep generative RKS. $Q, V$ and $K$ are linear transformations of input, $X$. The generator generates spectral frequency ($\Omega$) from an implicit spectral distribution. Using RKS (Eq. 2) we create the feature map $\phi_{gen}$ (Eq. 8). The numerator of the output is calculated as $\phi_{gen}(Q)(\phi_{gen}(K)V)$ while the denominator is $\phi_{gen}(q_i)^T \sum_{j'=1}^{L} \phi_{gen}(k_{j'})$ making attention linear in sequence length $L$.

| Model | Space Complexity | Time Complexity |
|---|---|---|
| Softmax Transformer | $\mathcal{O}(L^2(1 + d_q/L))$ | $\mathcal{O}(L^2 d_q)$ |
| Performer | $\mathcal{O}(L(d_q + M + Md_q/L))$ | $\mathcal{O}(LMd_q)$ |
| LinearElu | $\mathcal{O}(L(d_q + d_q^2/L))$ | $\mathcal{O}(Ld_q^2)$ |
| GMM-RKS | $\mathcal{O}(L(d_q + C(d_q^2/L + Md_q/L + M)))$ | $\mathcal{O}(MC(d_q^2 + Ld_q))$ |
| GMM-PRF | $\mathcal{O}(L(d_q + CMd_q/L + CM))$ | $\mathcal{O}(LCMd_q)$ |
| FASTFOOD(RKS/PRF) | $\mathcal{O}(L(d_q + M + Md_q/L))$ | $\mathcal{O}(LMd_q)$ |
| GENERATIVE(RKS/PRF) | $\mathcal{O}(L(d_q + d_q^2/L + Md_q/L + M))$ | $\mathcal{O}(M(d_q^2 + Ld_q))$ |

Table 1: Space and time complexity of self-attention kernel of KL-TRANSFORMERs compared with Softmax Transformer (Vaswani et al., 2017), Performer (Choromanski et al., 2021b), and LinearElu (Katharopoulos et al., 2020). $L$ refers to length of context, $d_q = d_v$ is the query/key/value dimension, while $M$ is the number of samples (where applicable).

$$\kappa(q,k) := \mathbb{E}_{\omega \sim \psi}[e^{(\omega^T q - \|q\|^2)} e^{(\omega^T k - \|k\|^2)}], \quad PRF(x, \Omega) := \frac{e^{-\|x\|^2}}{\sqrt{M}}[e^{\omega_1^T x}, ..., e^{\omega_M^T x}] \tag{9}$$

$$\phi_{\text{GMM}-\text{PRF}}(x) := PRF(x, (\omega_{c,1}, ..., \omega_{C,M})), \quad \omega_{c,m} = \Sigma_c n_m + \mu_c, \quad n_m \sim \mathcal{N}(\mathbf{0}, \mathbf{I}) \tag{10}$$

Similarly we redefine the other two methods as:

$$\phi_{\text{GENERATIVE}-\text{PRF}}(x) := PRF(x, \Omega), \quad \omega_m = g(n_m), \quad n_m \sim \rho_o(\cdot) \tag{11}$$

$$\phi_{\text{FASTFOOD}-\text{PRF}}(x) := PRF(x, V), \quad \text{where } V = \frac{1}{\sigma \sqrt{d_q}} SHG\Pi HB. \tag{12}$$

To the best of our knowledge we are the first to explore kernel learning with positive random features.

## 2.2 Analysis

In this section, we explore what can be said about the expressivity of the proposed linear time KL-TRANSFORMERs. While our understanding of the properties of Transformers is rudimentary, we would still like to know whether the known properties extend to our models. For example, it has been shown that Softmax Transformers and its sparse counterparts are Turing complete (Pérez et al., 2019; Zaheer et al., 2020).This raises the question as to whether the proposed linear KL-TRANSFORMERs are also Turing complete?

It is easy to see that the generalized kernel self-attention (Eq. 1) includes the softmax kernel and hence should satisfy the properties of Softmax Transformer. Interestingly, we can also show that this property holds for GMM-RKS Transformers with number of components $C = 1$, (for a more systematic definition, see Section A.1). More formally,

**Theorem 1:** *The class of* GMM-RKS *Transformers with positional embeddings is Turing complete.*

Proof is in the Appendix A. We also show that:

**Theorem 2:** *The class of* GMM-PRF *Transformers with positional embeddings is Turing complete.*

For a detailed proof, see Appendix A.

Since the sampling of $\omega$ in Equations 5 and 9 is random, we have some stochasticity in the attention layers of our model. We now show that the Mean Square Error (MSE) of the estimation can be reduced by reducing the eigenvalues of the learnt covariance matrix. In particular, we show that:

**Theorem 3:** *Let $\mu$ and $\Sigma = S^T S$ be the learnt mean an covariance matrices of the sampling distribution. Further let $q$ and $k$ be the parameters, and $p = k - q$ and $o = k + q$ be their vector difference and sum respectively, and m be the number of samples. The MSE of the linear approximations is given by:*

$$MSE_{\text{GMM-RKS}} = \frac{2}{m}\cos^2(\mu^T p)(1 - e^{-||S^T p||^2})^2 \tag{13}$$

$$MSE_{\text{GMM-PRF}} = \frac{1}{m}e^{-2(||q||^2 + ||k||^2 - \mu^T o)}(e^{2||S^T o||} - e^{||S^T o||}) \tag{14}$$

It is interesting to note that in this formulation, $MSE_{\text{GMM-RKS}}{}^2$ is bounded above by $\frac{2}{m}$ while no such bound can be established for $MSE_{\text{GMM-PRF}}$. For the detailed proof see Supplementary Section B

**Complexity Analysis:** While all of our models have linear complexity with respect to the context length $L$, differences still exist amongst the various methods. Notable, GMM-RKS and GENERATIVE have quadratic time and space complexity in the query size $d_q$. Both the FASTFOOD methods avoid this approximation, whereas GMM-PRF avoids this by the use of a diagonal covariance matrix. The complexities are listed in Table 1.

Another factor that controls timing is the sampling of $\Omega$. Sampling too frequently can lead to significant slowdowns whereas sampling too few times can lead to biased learning. For our experiments, we resample every 100 training iterations, although this can be changed. A detailed list of all hyperparameters along with implementation details are provided in Appendix C.

## 3 Experiments

### 3.1 Does kernel learning improve performance of fixed kernel methods on longer sequences?

Long Range Arena (*LRA*; Tay et al. 2021b) is a diverse benchmark for the purpose of evaluating the ability of sequence models to reason under long-context scenarios. It includes tasks that vary both in terms of the context length (ranging from $1K$ to $4K$ tokens) as well as the data modalities (including text and mathematical expressions). We evaluate the KL-TRANSFORMER architectures introduced in Section 2.1 on the *Text*, *Retrieval* and *ListOps* tasks from *LRA* which deal with sentiment prediction from IMDB Reviews, document similarity classification and pre-order arithmetic calculations respectively. Both *Text* and *Retrieval* use character level encodings, bringing their maximum length to 4000 tokens. The *ListOps* dataset is synthetically generated and has a fixed length of 2000 tokens. For more details on the tasks, we refer the interested reader to the original paper by Tay et al. 2021b.

**Setup:** To ensure a fair comparison, we closely follow the same data preprocessing, data split, model size and training procedure as in (Tay et al., 2021b). Within each task, a common configuration is used across all KL-TRANSFORMER

---

[2]For the proof of Theorem 3, we use the secific case of $C = 2$, $\mu_1 + \mu_2 = 0$ and $\Sigma_1 = \Sigma_2$ which is used in the experiments. Since Theorem 1 and 2 however, hold in general

| Model | Complexity | *ListOps* 2K | *Text* 4K | *Retrieval* 4K | **Avg.** |
|---|---|---|---|---|---|
| Random Predictor | NA | 10.00 | 50.00 | 50.00 | 36.67 |
| Baseline Models | | | | | |
| Softmax Trans. (Vaswani et al.) | $\mathcal{O}(L^2)$ | 36.38 | 64.27 | 57.46 | 52.70 |
| Synthesizer (Tay et al.) | $\mathcal{O}(L^2)$ | 36.50 | 61.68 | 54.67 | 50.95 |
| Sinkhorn (Tay et al.) | $\mathcal{O}((L/B)^2)$ | 34.20 | 61.20 | 53.83 | 49.74 |
| Sparse Trans. (Child et al.) | $\mathcal{O}(L\sqrt{L})$ | 35.78 | 63.58 | 59.59 | 52.98 |
| Reformer (Kitaev et al.) | $\mathcal{O}(L\log L)$ | 36.30 | 56.10 | 53.40 | 48.60 |
| Local Attention (Parmar et al.) | $\mathcal{O}(LK)$ | 15.95 | 52.98 | 53.39 | 40.77 |
| Longformer (Beltagy et al.) | $\mathcal{O}(LK)$ | 36.03 | 62.85 | 56.89 | 51.92 |
| Linformer (Wang et al.) | $\mathcal{O}(L)$ | 35.49 | 53.49 | 52.27 | 52.56 |
| Big Bird (Zaheer et al.) | $\mathcal{O}(LK)$ | 37.08 | 64.02 | 59.29 | 53.46 |
| LinearElu (Katharopoulos et al.) | $\mathcal{O}(L)$ | 17.15 | 65.90 | 53.09 | 45.38 |
| Performer (Choromanski et al.) | $\mathcal{O}(L)$ | 36.00 | 65.40 | 53.82 | 51.74 |
| Kernelized Transformers | | | | | |
| GMM-RKS (Eq. 5) | $\mathcal{O}(L)$ | 18.55 | 63.95 | 58.64 | 47.05 |
| FASTFOOD-RKS (Eq. 6) | $\mathcal{O}(L)$ | 18.65 | 65.67 | 61.92 | 48.75 |
| GENERATIVE-RKS (Eq. 8) | $\mathcal{O}(L)$ | 18.50 | **66.50** | 64.76 | 49.92 |
| GMM-PRF (Eqs. 9, 10 ) | $\mathcal{O}(L)$ | 36.96 | 62.64 | 65.27 | 54.96 |
| FASTFOOD-PRF (Eqs. 9, 12 ) | $\mathcal{O}(L)$ | 37.05 | 64.66 | **71.13** | **57.61** |
| GENERATIVE-PRF (Eqs. 9, 11 ) | $\mathcal{O}(L)$ | **37.42** | 62.90 | 69.81 | 56.71 |

Table 2: Experimental results on the *LRA* benchmark. We report accuracy on the test set, except for *Text* where validation set is used. The best model is in boldface and the second best is underlined if within 1% f the best. Accuracy scores for all baseline models are from Tay et al. (2021b). Here, $L$ refers to the sequence length, $K$ refers to the size of a local window and $B \ll L$ is a model specific parameter. For our models, accuracy is averaged over 100 runs.

models based on the configuration specified in the *LRA* code repository[3]. We outline the hyperparameters for all tasks in Table 6 in the Appendix.

**Results:** The results across all *LRA* tasks are summarized in Table 2. KL-TRANSFORMER variants that learn the kernel function directly from the data in an end-to-end manner outperform the baseline models by occupying both best and second-best performances. We find that KL-TRANSFORMERs based on PRFs tend to outperform their RKS counterparts which is also reflected on the average *LRA* score, with FASTFOOD-PRF being the best-performing model.

## 3.2 Trade-off between Accuracy and Efficiency

We benchmark the efficiency of each KL-TRANSFORMER in terms of peak memory usage and training speed and compare it against three baseline models from the *LRA* benchmark. Specifically, we compare against other efficient Transformer architectures that employ fixed kernel feature maps (e.g. LinearElu and Performer) as well as the Softmax Transformer which is one of the strongest baseline models (see Table 2). We conduct efficiency benchmarks on the two *LRA* tasks with sequence length equal to $4K$ in order to assess the efficiency of these methods in modelling tasks that require a long context (results for the other two datasets are included in the Appendix). Speed measurements (steps per second) refer to wall-clock training time (including overheads). In both cases experiments are conducted on 8 NVIDIA TITAN RTX GPUs. The comparison is illustrated in Figure 3. On the *Text* task, GENERATIVE-RKS is the best performing model, although it consumes more memory than the remaining KL-TRANSFORMER architectures (it is still more efficient than the Softmax Transformer). LinearElu consumes the least amount of memory, while GMM-RKS provides a trade-off between the two. In *Retrieval* the situation is much clearer, with FASTFOOD-PRF and GENERATIVE-PRF outperforming significantly other models in terms of accuracy while having very low memory

---

[3]https://github.com/google-research/long-range-arena

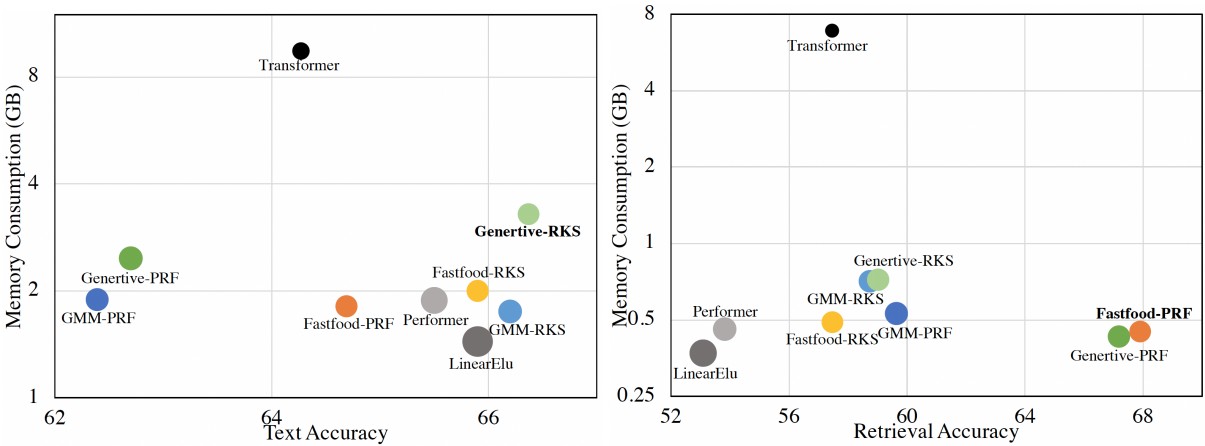

Figure 3: We compare the peak memory consumption (y-axis), performance (x-axis) and speed (denoted by area of circle) for the various KL-TRANSFORMER architectures on the two *LRA* tasks with sequence length equal to $4K$. Memory usage refers to average memory usage across GPUs and speed (steps per second) for each model is reported relative to the speed of Softmax Transformer (larger circles denote faster models). For a similar graph on the *ListOps* Task see Fig 8 in the Supplementary

consumption. The training speed of KL-TRANSFORMERs is of the same order of magnitude as Performers (as indicated by the area of each circle in Figure 3).

Lastly, in Figure 4, we report the peak memory consumption as the sequence length changes from $1K$ to $4K$ on the *Text* dataset. As expected, all our models have a linear increase in memory consumption with increasing sequence length, as opposed to the Softmax Transformer which has dramatic increase in memory consumption. Furthermore, Figure 7 in the Appendix reports the memory usage of each KL-TRANSFORMER across all datasets. We find that FASTFOOD-PRF and GENERATIVE-PRF are not only our best performing models on average, but they also consume the least memory among various KL-TRANSFORMERs across all datasets. Thus, among the models proposed in this paper, we can recommend FASTFOOD-PRF and GENERATIVE-PRF as the model that achieves the best accuracy with the least memory consumption.

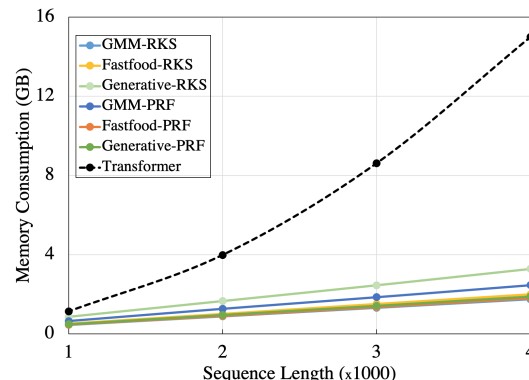

Figure 4: Memory vs sequence length

### 3.3 How do KL-TRANSFORMERs perform on short sequence tasks?

We compare the KL-TRANSFORMERs and Softmax Transformer in a common transfer learning setting. We adopt the setting of BERT-like models (Devlin et al., 2019; Liu et al., 2019), except that we have fewer layers and heads (see Table 8 for details) and pre-train all models (including Softmax Transformer) on the WikiText-103 dataset (Merity et al., 2016) using non-contextual WordPiece embeddings (Wu et al., 2016). Pre-trained models are then fine-tuned on the General Language Understanding Evaluation (*GLUE*) benchmark (Wang et al., 2019), a collection of resources for training and evaluating language understanding systems. All tasks in *GLUE* consist of either one or two sentences as input and can therefore be easily captured with a context of size $512$. Since the context length is rather short, the difference between training and inference time across the various models is minimal. Instead, the main goal of this task is to assess how do KL-TRANSFORMERs compare against Softmax Transformers on a set of tasks where the later have been established as the *de-facto* architecture.

The results on all downstream *GLUE* tasks are shown in Table 3. Crucially, we demonstrate that there is no significant loss in performance compared to Softmax Transformers when kernelized variants are used on short language modelling tasks. As illustrated in Table 3, KL-TRANSFORMERs perform on par with the Softmax Transformer.

| Model | SST2 (acc) | MRPC (acc/f1) | QQP (acc/f1) | MNLI-m/mm (acc/acc) | QNLI (acc) | WNLI (acc) | RTE (acc) | CoLA (MCor) |
|---|---|---|---|---|---|---|---|---|
| Softmax Trans. | 0.81 | 0.70/0.82 | 0.83/0.76 | 0.64/0.64 | 0.68 | 0.56 | 0.6 | 0.18 |
| FASTFOOD-RKS | 0.83 | 0.71/0.82 | 0.81/0.74 | 0.57/0.57 | 0.64 | 0.59 | 0.56 | 0.13 |
| GMM-RKS | 0.80 | 0.70/0.82 | 0.77/0.69 | 0.47/0.48 | 0.60 | 0.61 | 0.57 | 0.07 |
| GENERATIVE-RKS | 0.81 | 0.70/0.82 | 0.81/0.73 | 0.59/0.58 | 0.63 | 0.62 | 0.58 | 0.16 |
| FASTFOOD-PRF | 0.81 | 0.71/0.82 | 0.81/0.74 | 0.56/0.57 | 0.64 | 0.59 | 0.58 | 0.12 |
| GENERATIVE-PRF | 0.80 | 0.71/0.82 | 0.80/0.74 | 0.56/0.56 | 0.61 | 0.60 | 0.55 | 0.10 |
| GMM-PRF | 0.82 | 0.71/0.82 | 0.81/0.74 | 0.56/0.56 | 0.64 | 0.59 | 0.59 | 0.21 |

Table 3: Results on the GLUE benchmark after fine-tuning on respective tasks. KL-TRANSFORMERs continue to be competitive to Transformers even in short context problems.

| Task | Model | Max Egv. | Mean Egv. | Task | Model | Max Egv. | Mean Egv. |
|---|---|---|---|---|---|---|---|
| Text | GMM-RKS | 0.486 | 0.031 | Retrieval | GMM-RKS | 0.499 | 0.053 |
| | GMM-PRF | 0.096 | 0.025 | | GMM-PRF | 0.186 | 0.042 |

Table 4: Distribution of Eigenvalues for GMM-RKS and GMM-PRF models for the *Text* and *Retrieval* tasks. For a head by head distribution see Tables 9, 10, 11 and 12

## 4 Empirical Analysis

In section 3, we observed that our models compare favourably with other linear models, with Positive Random Features (PRF) based models usually doing better that the ones based on Random Kitchen Sinks (RKS) (although RKS does better on the *Text* task). In this section, we want to compare these linear kernels in terms of their empirical variance and how well they deal with sparsity. In Theorem 3, we already noted that the MSE for both GMM-RKS and GMM-PRF models decreases with decrease in eigenvalues of the covariance matrices, making it useful to learn such a matrix. We also noted that the variance in the output of GMM-RKS is bounded, and thus it should not face issues with sparse datasets when approximating the softmax kernel as show in Choromanski et al. (2021b)). We test for these results empirically in subsections 4.1 and 4.2 respectively. We then conclude in 4.3 with a discussion on which of our proposed models is best to use in a given scenario.

### 4.1 Comparison of variance

Looking at the eigenvalues of the covariance matrices of our final trained models[4] in Table 4, we find that GMM-RKS and GMM-PRF models trained on the *Text* and *Retrieval* tasks indeed learn to have eigenvalues significantly smaller than 1 (which would be the case for non-learnable co-variances).

While lower eigenvalues reduce the variance in our models significantly, it does not get completely nullified. To understand how much stochastic remains in our models, and also to compare RKS with PRF in regards to their variance, we look at the final outputs[5] the models produce when repeatedly run on the same example. We record the output produced by the model for 100 runs on each datapoint, and calculate the following 3 metrics to quantify the variance:

1. **Relative Standard Deviation (RSD):** RSD is a standardised measure of dispersion (see Currie & Svehla (1994), def 3.9), defined as the ratio of the standard deviation of a set of observations to the absolute value of their mean. The ratio ensures that the measure is invariant to scaling (eg. multiplying the penultimate layer of the model by a constant factor does not affect RSD) while the absolute of the mean ensures that opposing classes don't cancel each other out.

---

[4]We only report eigenvalues for the GMM models since covariance matrices are not well defined in the Generator and FastFood. Also, the *ListOps* task is left out of the entire analysis because its multi-class nature makes the analysis complex

[5]While the theory makes a claim about the output of each attention head, evaluating every head at every layer would give us a large number of values to analyse. The output is considered before aplying the final sigmoid

2. **Prediction Inconsistency (PI):** While RSD quantifies the variance in the model's continuous output, in practice, only the sign of the output is of interest as that decides the predicted class. As a way to quantify stochasticity in the discrete output, we count the number of times the output does not have the majority label, i.e., if the output is positive $x$ times, then we have $PI = \min(x, 100 - x)$. Alternately, it can be seen as the failure rate if we treat the majority prediction as the true class.

3. **Accuracy Gain with Voting (AGV):** Since the final output has a propensity to change its sign, we can get a more robust prediction at inference time by running the model multiple times and considering the class it predicts more times as its output, instead of taking the prediction from a single inference step. Doing so, we are likely to get closer to the mean prediction (by the Law of Large Numbers, see Révész et al. (2014)), and we get accuracy scores which are slightly higher than those reported in Table 2, and we call this value the *Voting Accuracy (VA)*. AGV is then defined as the ratio between the voting accuracy and the regular accuracy.

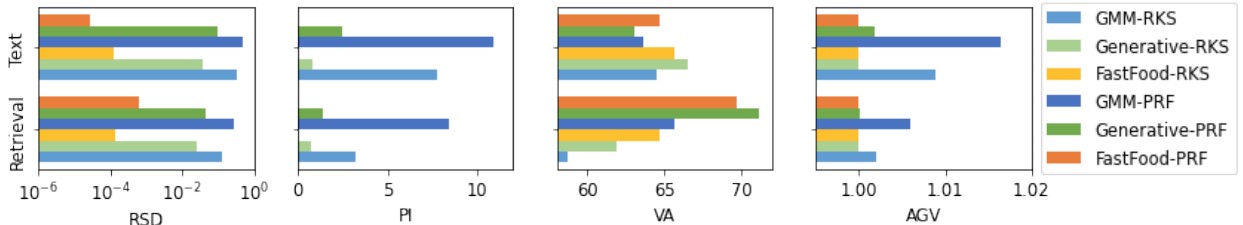

Figure 5: Values of variance metrics Relative Standard Deviation (RSD), Prediction Inconsistence(PI), Average Gain with Voting (AGV) as well as Voting Accuracy (VA) on the *Text* and *Retrieval* tasks.

The above metrics, in addition to the *Voting Accuracy*, are calculated over all instances in the validation and test set for *Text* and *Retrieval* tasks respectively, and plotted in Fig. 5. We note that our 3 metrics are all highly correlated with each other ($R^2$ values 0.95, 0.98 and 0.89 for RSD-AGV, RSD-PI and AGV-PI respectively, see supplementary Fig 9, 10).

We notice that our RKS based models, in general, have lower variance as compared to our PRF models. We also note that Generator and FastFood are able to further reduce the variance in the output, possibly due to additional learnable parameters in Generator and fewer sources of randomness in FastFood. We also notice that all models have greater variance in the Text task, and it is possible that this is related to the better performance of RKS based models in this task, and their inherent lower variance helps them outperform their PRF counterparts.

## 4.2 Effectiveness on Sparse Datasets

Choromanski et al. (2021b) demonstrated that if Random Kitchen Sinks are used to estimate the softmax-kernel, the MSE tends to $\infty$ as the raw softmax scores tend to $0$. This makes it particularly hard to deal with sparse datasets where multiple positions need near-zero attention.

Thus, in order to test how sparsity in the dataset affects our models, we design a synthetic experiment where the model is fed a sequence of ordered pairs, where the first element, which we call *score*, takes a value of $-1$ or $1$, and the second element, which we call *relevance*, takes a value of $0$ or $1$. Looking at the sequence, the model must predict the sum of the scores at all positions with relevance $1$. Note that any position with relevance $= 0$ does not contribute to the final answer, and therefore, does not need to be attended to. We construct the inputs by sampling the relevance of each position from Bernoulli($p$). Thus, the sparsity can be controlled by changing $p$ (for a detailed description of the task, see Supplementary).

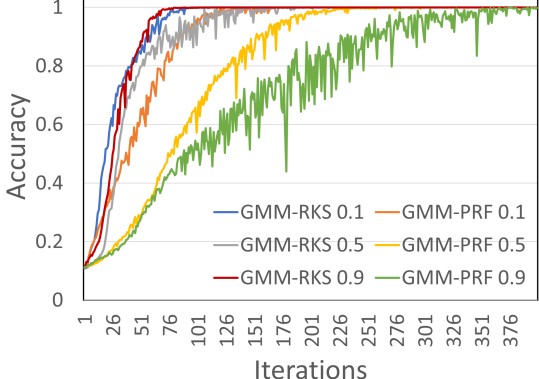

Figure 6: Learning curves for our synthetic experiment. The number after the model name is inversely proportional to sparsity in the dataset

Figure 6 shows the learning curves for various sparsity levels in our synthetic experiment. Due to the simple nature of the task, all models eventually reach 100% accuracy. We observe that

the convergence time of GMM-RKS remains more or less unchanged with increasing sparsity, while for GMM-PRF the model converges slower as sparsity decreases. We believe that the slower convergence of GMM-PRF is correlated to the variance in its output, which leads to variance in gradients. Observing the gradients propagated to the classifier layer (See Table 14 in Supplementary), we find that this is indeed the case, and not only is the variance in the gradients higher for GMM-PRF, but also the mean is higher, making it take bigger steps in an uncertain direction. This result provides us an insight into which models to use under which circumstances.

### 4.3 Which Model to Use?

PRF outperforms RKS is terms of accuracy in most of our experiments, especially if we are able to run multiple inference steps. Therefore, in general, we recommend using PRF over RKS. However, if consistency of predictions is important, or rapid training is required on a not-so-sparse task, one may consider RKS as well.

Finally, we observe that Generator and Fastfood methods always outperform the vanilla linearisations. Between them, Fastfood can be significantly faster if properly implemented on GPU. However, generators may perform better if a large amount of data is available since they provide a greater amount of flexibility.

## 5 Related Work

### 5.1 Efficient Transformers

A wide variety of approaches belong to the class of efficient Transformers (Tay et al., 2020b). We survey them below:

**Sparse models:** Memory Compressed Transformer (Liu et al., 2018) uses a convolution kernel (of size $K$) to sub-sample keys and values reducing the complexity to $\mathcal{O}(L^2 K^{-1})$. Inspired by the notion of sparsity, Child et al. (2019) introduced sparse factorizations of the attention matrix to reduce the overall complexity to $\mathcal{O}(L\sqrt{L})$. Subsequently, Roy et al. (2020) proposed the Routing Transformer which employs $K$-means clustering to learn dynamic sparse attention regions, achieving an overall complexity of $\mathcal{O}(L\sqrt{L})$. Recently, Sun et al. (2022) proposed yet another sparse $\mathcal{O}(L\sqrt{L})$ approach that learns which bucket each query/key is to be placed in based on final attention values. Sparsity has also been achieved by efficiently subsampling queries and/or keys (Chen et al., 2021). Kitaev et al. (2020) proposed the Reformer, which reduces complexity to $\mathcal{O}(L \log L)$ by using locality-sensitive hashing to group together similar symbols. Ye et al. (2019) also proposed a $\mathcal{O}(L \log L)$ algorithm using binary partitions of data. There also exist other works that mainly focus on memory reduction (Liu et al., 2018; Tay et al., 2020a; Gupta et al., 2021). While these methods are faster than Softmax Transformers, their asymptotic time complexity remains quadratic. Further approaches attempt to minimize the constants involved in the quadratic attention, but keep the same assymptotic complexity (Dutta et al., 2021; Chen et al., 2022).

**Local or Global Attention:** Parmar et al. (2018) was one of the first local attention models that achieved $\mathcal{O}(L)$ complexity in both time and space, by using local attention over a constant length-context. A similar local attention based method was also utilized by Zhang et al. (2021) and Liu et al. (2022), the latter of which added special tokens at the start of each local attention block, which attend globally. Another set of methods proposed to approximate the global attention by replacing the softmax to allow changing the order of matrix multiplication, making the calculation of the $QKV$ product linear in the context length (Yorsh et al., 2021; Hua et al., 2022; Qin et al., 2022). Linearised global attention is also used by Performer (Choromanski et al., 2021b), which has been built upon by other works. Chen (2021) and Luo et al. (2021) both try to improve the performance of Performer by incorporating relative positional embeddings which they show makes it strictly stronger in terms of representability. Choromanski et al. (2021a) tries to combine RKS and PRF attentions by a learnable weight parameter to get the best of both worlds. Both these methods can be applied as is to our work. Schlag et al. (2021a) take a different view on this, claiming that having stochasticity hampers performance, and propose a deterministic feature map to avoid this.

**Multiple Types of Attention:** Beltagy et al. (2020) proposed a $\mathcal{O}(L)$ method that combines the above two approaches by using local sliding windows as well as global attention components. Zaheer et al. (2020) proposed Big Bird, another sparse attention mechanism which combines global attention, random attention and local attention to reduce the complexity to $\mathcal{O}(L)$. A similar construction was previously used by Ainslie et al. (2020). More recently the combination of global and local attention has been utilized by Zhu et al. (2021) and Nguyen et al. (2021). Further, Xiong et al. (2021) adapted the Nyström method to approximate standard self-attention with $\mathcal{O}(L)$ complexity. Lu et al. (2021) also makes

use of a similar near-field and far-field attention mechanism, where the far field attention is calculated via a low rank approximation using $tanh$ and $ReLU$ non-linearities. A different line of work has attempted to limit the number of key-value pairs that are to be attended to, by attempting to summarise the variable length context with a fixed number of memory-cells (Zhang & Cai, 2022; Ma et al., 2021). While some of these more nuanced approaches outperform KL-TRANSFORMER, most of their innovations are orthogonal to ours, and none of these approaches explore kernel learning within the attention mechanism of Transformers.

## 5.2   Kernel Learning

While kernel methods have long been used to solve non-linear statistical problems, they traditionally scaled poorly with the number of data points thereby limiting their applicability on large datasets (Vapnik et al., 1997; Cortes & Vapnik, 1995; Schölkopf et al., 1998; Schölkopf & Smola, 2001; Hofmann et al., 2008). Prior to RKS, several kernel approximation techniques have been proposed to improve the scalability of kernel methods, including greedy basis selection techniques (Smola & Schökopf, 2000), divide-and-conquer approaches (Hsieh et al., 2014; Zhang et al., 2013; Liu et al., 2020), non-stationary spectral kernels (Remes et al., 2017), generalized spectral kernels (Samo & Roberts, 2015) as well as Nyström methods (Williams & Seeger, 2001).

The method of Random Kitchen Sinks has been revisited several times, either to improve the approximation quality (Yu et al., 2016b; Choromanski et al., 2017; Li, 2017; Avron et al., 2016; Lyu, 2017), reduce the time and memory complexity (Le et al., 2013; Choromanski & Sindhwani, 2016; Feng et al., 2015; Dao et al., 2017) or analyze theoretically the risk and generalization properties of the algorithm (Sutherland & Schneider, 2015; Sun et al., 2018; Li et al., 2019b). A systematic survey of random feature methods for approximating kernel functions can be found in (Liu et al., 2021).

Lastly, there exist a class of methods that extend the RKS framework to enable kernel learning. Representative approaches involve either a one-stage (Yang et al., 2015; Fang et al., 2020) or a two-stage procedure (Sinha & Duchi, 2016; Li et al., 2019a; Bullins et al., 2018; Shen et al., 2019). Two-stage approaches involve an intermediate step in which the problem of kernel-alignment is solved (Cristianini et al., 2002). However, solving the kernel-alignment problem requires accessed to labeled data which is not available in this case, as inputs to the kernel learning algorithm are the intermediate representations of the input sequence.

## 6   Conclusion

In this paper, we bridged the gap between advances in kernel learning and efficient Transformers by proposing several kernel learning methods for Transformers that increase the expressiveness of Transformers while keeping the computational complexity linear in sequence length. We showed that our proposed KL-TRANSFORMER are Turing-complete and can control their variance. Experimentally our proposed models perform on par with, and possibly exceed the performance of existing efficient transformer architectures on long context tasks without falling behind on short context tasks. We also found that for some datasets such as *ListOps*, RKS based models tend to fall short of their PRF counterparts. Our experiments further demonstrate that the memory consumption of our models scales linearly with the sequence length.

## Ethical Considerations

Our work is on making Transformers computationally efficient without losing expressiveness. Our models were evaluated on publicly available benchmark datasets. The datasets used in our work do not contain sensitive information to the best of our knowledge.

**Reproducibility:** We plan to open source the entire code of the KL-TRANSFORMER framework (including the implementation of all models as well as the code for replicating all of our experiments) before the camera ready version of the paper. As part of this submission, we include code for all the methods proposed by us along with instructions on how to reproduce results. A detailed description of all hyperparameters (for both *LRA* as well as *GLUE* benchmarks) has been included in Appendix C. Finally, regarding our theoretical contributions, we present a detailed theoretical analysis in Appendix A.

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

# Learning The Transformer Kernel – Appendix

## A Detailed proof of Theorems

### A.1 Definitions

**Transformer:** A transformer consists of an Encoder and a Decoder, which in turn consist of several encoder and decoder layers respectively. A single encoder layer consists of an attention layer($Att$) and a 2 layer feed-forward neural network($O$) :

$$\mathbf{a}_i = Att(W_q \mathbf{x}_i, W_k \mathbf{X}, W_v \mathbf{X}) + \mathbf{x}_i \tag{15}$$

$$\mathbf{z}_i = O(\mathbf{a}_i) + \mathbf{a}_i \tag{16}$$

In our case, the feed-forward neural network uses perceptron activations(ie $f_{perc}(x) = 1$ iff $x > 0$ and 0 otherwise) and the attention is gaussian(discussed in detail later). The final layer of the encoder is followed by a couple of two layer output neural networks, which produce the Encoder Key($\mathbf{K_e}$) and Encoder Value($\mathbf{V_e}$) to be used by the decoder. In our proof, we assume these to have ReLU activation($f_{ReLU}(x) = max(0, x)$)

The decoder layers are similar to the encoder layer except for an additional cross attention layer which attends to the encoder output:

$$\mathbf{p}_i = Att(W_q \mathbf{y}_i, W_k \mathbf{Y}, W_v \mathbf{Y}) + \mathbf{y}_i \tag{17}$$

$$\mathbf{a}_i = Att(W_q' \mathbf{p}_i, \mathbf{K_e}, \mathbf{V_e}) + \mathbf{y}_i \tag{18}$$

$$\mathbf{z}_i = O(\mathbf{a}_i) + \mathbf{a}_i \tag{19}$$

Unlike the encoder, the decoder self attention if Eq. 17 can only attend to previous position. After the final layer we have a two layer feed-forward neural network with ReLU activation to produce the output. The decoder is initialised with a special *seed vector* and is repeatedly applied with the right shifted output of the last application as the input of the current application, until some termination condition is fulfilled.

Both the encoder and decoder can further use *position embeddings*, which have the same dimension as the output of each layer, and are added to the input prior to the first layer. These help in establishing the order of the input

Since the output of any unit of a layer is independent of values to its right, these do not change with time and can be cached. The output of the final layer of the rightmost cell can therefore be regarded as the model state encoding($v$)

**Turing Machine:** A Turing Machine is an abstract construct which consists of a right infinite tape and a read-write head. Each cell of the tape can hold one of many symbols from a predefined alphabet $\Sigma$ which includes a special blank symbol $b$. Additionally, the read-write head can be in one of many possible states within the state-space $Q$ which includes a special initial state $q_{init}$ and a subset of final states $F$.

Initially, the tape contains the input followed by an infinite number of blank symbols, while the head starts off in the last non-blank cell. In each step, the head executes in accordance with a transition function $T(s^{(i)}, q^{(i)}) = (v^{(i)}, q^{(i+1)}, m^{(i)})$ , ie, based on the symbol currently under the head and the current state, it decides the symbol it wants to overwrite the current symbol with, the state it will be in the next step and the direction it wants to move, which can be either left($-1$) or right($1$). We assume that the transition function already makes sure that the head never moves left from the leftmost cell.

For the purpose of our proof, we additionally define $c^{(i)}$ as the index of the cell to which the head currently points, $\ell(i)$, which represents the step number when the head last pointed to the current cell, ie $\ell(i) = \max\{j|c^{(j)} = c^{(i)}\}$. In the special case where the current cell is being visited for the first time, we have $\ell(i) = i - 1$

### A.2 The Proof

In this section, we provide a general proof which works for both **Theorem 1** and **Theorem 2** in the paper. This is possible since the construction only makes use of the dual of the kernel functions used, i.e. the gaussian. The fact that

both kernel functions map to the gaussian is shown in lemma S.2 (Sec A.6)

Our proof is based on the similar proof in Pérez et al. (2019). Any symbols not explicitly defined have same meanings from that paper. We begin the proof by defining our model encoding($\mathbf{v}$):

$$\begin{aligned} \mathbf{v} = [&\mathbf{q}_1, \mathbf{s}_1, x_1, x_2 \\ &\mathbf{q}_2, \mathbf{s}_2, x_3, x_4, x_5, x_6, \\ &x_7, \mathbf{s}_4, x_8, \\ &x_9, x_{10}, x_{11}, x_{12}] \end{aligned} \tag{20}$$

where $q_i \in \mathbb{Q}^{|Q|}$, $s_i \in \mathbb{Q}^{|\Sigma|}$, and $x_i \in \mathbb{Q}$, giving a total model size of $2|Q| + 3|\Sigma| + 14$. Hereafter, $[\![x]\!]$ represents the one-hot encoding for the state $x$ or symbol $x$ depending on the position it is being used in. $\mathbf{0_q}$ represents all 0's in a state field, and represents the $q_{copy}$ state discussed later, while $\mathbf{0_s}$ represents all 0's in a symbol field, and represents the blank symbol. Further, $\beta^{(i)} = \min(i, n)$ where $n$ is the size of the encoder and $\alpha^{(i)}$ represents the symbol at position $\beta^{(i)}$ in the encoder. We assume that atleast the last cell of the encoder contains a blank symbol.

This differs from Pérez et al. (2019) in the addition of a fourth scalar in the first group, in which we intend to store the current position $c^{(i)}$ of the head.

Our invariant is that $\mathbf{y}_i$ the output from the decoder at timestep $i$, stores:

1. The current state of the Turing Machine($q^{(i)}$)

2. The symbol under the head($s^{(i)}$)

3. The direction of movement of the head in the previous timestep ($m^{(i-1)}$)

4. The current position of the head($c^{(i)}$)

In all, we get $\mathbf{y}_i = [[\![q^{(i)}]\!], [\![s^{(i)}]\!], m^{(i-1)}, c^{(i)}, 0, \dots, 0]$

**Positional Embeddings:** The last group $(x_9, x_{10}, x_{11}, x_{12})$ is dedicated to the positional embeddings, which are given as$(1, i, \frac{1}{i}, \frac{1}{i^2})$ These same embeddings are added on both the Encoder and Decoder side.

**Encoder**: The encoder consists of a single layer. It gets as input the symbol at position $i$ and the positional embeddings, ie $input_i = [\mathbf{0}_q, \mathbf{0}_s, 0, 0, \mathbf{0}_q, , [\![s^{(i)}]\!], 0, 0, 0, 0, i, \mathbf{0}_s, 0, 1, i, \frac{1}{i}, \frac{1}{i^2}]$ which has a trivial attention layer(ie, one that outputs all zeroes) and a feed forward layer which separates the positional embeddings from the symbols, giving $\mathbf{k}_i^e = [0, \dots, 0, i, -1, 0, 0]$ and $\mathbf{v}_i^e = [\mathbf{0}_q, \mathbf{0}_s, 0, 0, \mathbf{0}_q, , [\![s^{(i)}]\!], 0, 0, 0, 0, i, \mathbf{0}_s, 0, 0, 0, 0, 0]$.

**Decoder Layer 1**: The first layer of the decoder calculates the next state, the symbol to be written and the direction of movement of the head. This includes 2 cases:

1. Initially, the Decoder starts off with in the state $q_{copy}$. While the state is still $q_{copy}$, the head writes the symbol at the $i^{th}$ position in the encoder and moves right, until a blank symbol is seen. Once a blank symbol is reached, the tape rewrites the blank symbol, moves left and the state changes to $q_{init}$.

2. Once we move into $q_{init}$, the output is fully defined by the current state and symbol under the head.

To facilitate the first case, we make use of the cross attention layer, to get

$$\begin{aligned} Att(\mathbf{q}, \mathbf{K^e}, \mathbf{V^e}) = [&0, \dots, 0, \\ &\mathbf{0}_q, [\![\alpha^{(i)}]\!], 0, 0, 0, 0, \\ &\beta^{(i)}, \mathbf{0_s}, 0, \\ &0, 0, 0, 0] \\ = &\mathbf{v}_{\beta(i)}^e \end{aligned} \tag{21}$$

The details of this process are explained in lemma S.1(see sec. A.4) Adding in the residual connection, we have:

$$
\begin{aligned}
\mathbf{a_i^1} = [&[\![q^{(i)}]\!], [\![s^{(i)}]\!], m^{(i-1)}, c^{(i)} \\
& \mathbf{0}_q, [\![\alpha^{(i)}]\!], 0, 0, 0, 0, \\
& \beta^{(i)}, \mathbf{0_s}, 0, \\
& 1, i+1, \frac{1}{(i+1)}, \frac{1}{(i+1)^2}]
\end{aligned}
\tag{22}
$$

Hereafter, we make use of the feed-forward layer to get:

$$
\begin{aligned}
O(\mathbf{a_i^1}) = [&-[\![q^{(i)}]\!], -[\![s^{(i)}]\!], -m^{(i-1)}, m^{(i)}, \\
& [\![q^{(i+1)}]\!], [\![\bar{v}^{(i)}]\!], m^{(i)}, m^{(i-1)}, 0, 0, \\
& 0, \dots, 0 \\
& 0, \dots, 0]
\end{aligned}
\tag{23}
$$

If the state is $q_{init}$ then we set $[\![\bar{v}^{(i)}]\!] = \mathbf{0}_s$, else we have $[\![\bar{v}^{(i)}]\!] = [\![v^{(i)}]\!]$. Note that this gives us $[\![\bar{v}^{(i)}]\!] + [\![\alpha^{(i)}]\!] = [\![v^{(i)}]\!]$.

To get all the required values, we first project$[\![q^{(i)}]\!]$ and $[\![s^{(i)}]\!]$ to a one-hot encoding of $Q \times \Sigma$. from there, we can calculate all the required values in a look-up table fashion. if the state is $q_{init}$ then we set $[\![v^{(i)}]\!] = \mathbf{0}_s$

The final output of this layer is then:

$$
\begin{aligned}
\mathbf{z_i^1} = [&0, \dots, 0, c^{(i+1)} \\
& [\![q^{(i+1)}]\!], [\![v^{(i)}]\!], m^{(i)}, m^{(i-1)}, 0, 0, \\
& \beta^{(i)}, \mathbf{0_s}, 0, \\
& 1, i+1, \frac{1}{(i+1)}, \frac{1}{(i+1)^2}]
\end{aligned}
\tag{24}
$$

**Decoder Layer 2:** In this layer we calculate the symbol under the head in the next timestep. In order to do so, we first use the self attention layer to calculate $[\![v^{(\ell(i+1))}]\!]$ and $(\ell(i+1))$(For details, see sec A.7.):

$$
\begin{aligned}
Att(W_q^2 \mathbf{z}_i^2, W_k^2 \mathbf{Z}^2, W_v^2 \mathbf{Z}^2) = [&0, \dots, 0, \\
& 0, \dots, 0, \\
& 0, [\![v^{(\ell(i+1))}]\!], (\ell(i+1)), \\
& 0, 0, 0, 0]
\end{aligned}
\tag{25}
$$

Adding the residual layer, we have

$$
\begin{aligned}
\mathbf{a_i^2} = [&0, \dots, 0, c^{(i+1)} \\
& [\![q^{(i+1)}]\!], [\![v^{(i)}]\!], m^{(i)}, m^{(i-1)}, 0, 0, \\
& \beta^{(i)}, [\![v^{(\ell(i+1))}]\!], (\ell(i+1)), \\
& 1, i+1, \frac{1}{(i+1)}, \frac{1}{(i+1)^2}]
\end{aligned}
\tag{26}
$$

The feed-forward layer then gives $O_2(\mathbf{a_i^2}) = [[\![q^{(i+1)}]\!], [\![v^{(\ell(i+1))}]\!] - f_{perc}((\ell(i+1)+2-(i+1)), m^{(i-1)}, 0, -M, \dots - M]$ where M is a large negative value. The perceptron function in the $\mathbf{s}_1$ is added positionwise, and is 0 unless $\ell(i+1) = i$. In this special case, it makes $\mathbf{s}_1$ contain only 0 or $-1$ which is converted into $\mathbf{0}_s$ by the ReLU activation in the output MLP. The same is also true for every field after the first 4, where we add a large negative value to make the ReLU output 0.

### A.3 The Attention Mechanism

The attention mechanism in the Gaussian kernel is defined as follows:

$$Attn(\mathbf{Q}, \mathbf{K}, \mathbf{V}) = \mathbf{V}\big(ColNorm\big(\Phi(\mathbf{Q})^T\Phi(\mathbf{K})\big)\big) \tag{27}$$

where $\Phi$ is $\Phi_{RKS}$ (Eq 34) for Theorem 1 and $\Phi_{PRF}$ (Eq 33) for Theorem 2, and $\omega_i$ is sampled from a gaussian with zero mean and diagonal covariance. However, for the proof construction, we use a hard version of this attention mechanism, and limit ourselves to the standard gaussian for $\omega$(since the mean and sigma is learnable, this can always be achieved). To begin with, we replace the kernels with their common dual, using lemma S.2 (sec A.6). In our construction, we do not require learnable means and variances, so we fix them to be $0_{d_q}$ and $\mathbb{I}$ hereafter:

$$Attn(\mathbf{Q}, \mathbf{K}, \mathbf{V}) = \sum_{i=0}^{h-1} \mathbf{W}_O^i \mathbf{V}\Big(ColNorm\Big[\!\Big[e^{-\frac{||\mathbf{q}_l - \mathbf{k}_m||^2}{2}}\Big]\!\Big]_{l=0,m=0}^{d-1,d-1}\Big) \tag{28}$$

where $[\![f(l,m)]\!]_{l=0,m=0}^{\alpha,\beta}$ denotes an $\alpha \times \beta$ matrix whose $(l,m)^{th}$ entry is $f(l,m)$, $ColNorm(\mathbf{X})$ indicates the matrix $\mathbf{X}$ with its columns normalised to and $d$ is the dimension of the query/key vector.

While this definition does not seem to allow multiplying the exponent, one must remember that the query and key matrices are calculated using projection matrices, and any required scalar factor can be incorporated into them. Therefore, we define hard gaussian attention as:

$$score(\mathbf{u}, \mathbf{v}) = -||\mathbf{u} - \mathbf{v}||^2 \tag{29}$$

Hard attention is them computed as

$$Att(\mathbf{q_i}, \mathbf{K}, \mathbf{V}) = \frac{\sum_{j=0}^{n-1} \mathbb{I}[score(\mathbf{q_i}, \mathbf{k_j}) = (max_{j'} score(\mathbf{q_i}, \mathbf{k_{j'}}))]\mathbf{v_j}}{\sum_{j=0}^{n-1} \mathbb{I}[score(\mathbf{q_i}, \mathbf{k_j}) = max_{j'}(score(\mathbf{q_i}, \mathbf{k_{j'}}))]} \tag{30}$$

Here $\mathbb{I}$ in the indicator function.

### A.4 Lemma S.1

### A.5 Statement

Given

$$\begin{aligned}
\mathbf{q} &= [\_\_, \ldots, \_\_, 1, i, \_\_, \_\_] \\
\mathbf{k}_j^e &= [0, \ldots, 0, \\
&\quad\quad 0, \ldots, 0, \\
&\quad\quad 0, \ldots 0, \\
&\quad\quad j, -1, 0, 0] \\
\mathbf{v}_j^e &= [0, \ldots, 0, \\
&\quad\quad \mathbf{0}_q, [\![s^{(j)}]\!], 0, 0, 0, 0, \\
&\quad\quad j, \mathbf{0_s}, 0, \\
&\quad\quad 0, 0, 0, 0]
\end{aligned} \tag{31}$$

For $j \in \{0, \ldots, n\}$, we need a construction that gives

$$\begin{aligned}
Att(\mathbf{q}, \mathbf{K^e}, \mathbf{V^e}) &= [0, \ldots, 0, \\
&\quad\quad \mathbf{0}_q, [\![\alpha^{(i)}]\!], 0, 0, 0, 0, \\
&\quad\quad \beta^{(i)}, \mathbf{0_s}, 0, \\
&\quad\quad 0, 0, 0, 0] \\
&= \mathbf{v}_{\beta(j)}^e
\end{aligned} \tag{32}$$

### A.5.1 Proof

Note that while the key and value comes from the encoder, and is therefore fixed, the query comes from the decoder and thus can be projected as we please. It is easy to construct a projection matrix that gives $W_Q \mathbf{q} = [0, \ldots, i, -1, 0, 0]$. Then we have $score(\mathbf{q}, \mathbf{k}_{j'}) = -||i = j||^2 = -(i - j)^2$, whose maxima on $j'$ is unique and occurs at $i = \beta^{(j)}$. Thus, we have $Att(\mathbf{q}, \mathbf{K^e}, \mathbf{V^e}) = \mathbf{v}_{\beta^{(j)}}^e$, which is exactly what we wanted.

## A.6 Lemma S.2

### A.6.1 Statement

Let

$$\Phi_{PRF}(\mathbf{x}) = [\exp(-||\mathbf{x}|| + \omega_0^T \mathbf{x}), \ldots, \exp(-||\mathbf{x}|| + \omega_{k-1}^T \mathbf{x})] \tag{33}$$

$$\Phi_{RKS}(\mathbf{x}) = \sqrt{\frac{2^m}{k}} [\cos(\omega_1^T x), \ldots, \cos(\omega_k^T x), \sin(\omega_1^T x), \ldots, \sin(\omega_k^T x)] \tag{34}$$

We want to show that if $\omega \sim \mathcal{N}(0, \mathbf{I})$ then the kernels as defined above corresponds to the *GMM kernel*, ie.

$$\Phi_{RKS}(\mathbf{X})\Phi_{RKS}(\mathbf{Y}) \approx \Phi_{PRF}(\mathbf{X})\Phi_{PRF}(\mathbf{Y}) \approx e^{-\frac{||\mathbf{x}-\mathbf{y}||^2}{2}} \tag{35}$$

### A.6.2 Proof

**Positive Random Features:** The proof actually follows from a trivial extension of Lemma 1 in (Choromanski et al., 2021b), but we present it here end to end for the convenience of the reader.

First we observe that

$$\begin{aligned}
e^{-\frac{||\mathbf{x}-\mathbf{y}||^2}{2}} &= e^{-\frac{||\mathbf{x}||^2 - 2\mathbf{x^T y} + ||\mathbf{y}||^2}{2}} \\
&= e^{-\frac{2*||\mathbf{x}||^2 - ||\mathbf{x}||^2 - 2\mathbf{x^T y} + 2*||\mathbf{y}||^2 - ||\mathbf{y}||^2}{2}} \\
&= e^{-||\mathbf{x}||^2} e^{\frac{||\mathbf{x}+\mathbf{y}||^2}{2}} e^{-||\mathbf{y}||^2}
\end{aligned} \tag{36}$$

Next we leverage the fact that $(2\pi)^{-d_q/2} \int e^{-\frac{||\omega - \mathbf{c}||^2}{2}} d\omega = 1$ in to evaluate the second factor above:

$$\begin{aligned}
e^{\frac{||\mathbf{x}+\mathbf{y}||^2}{2}} &= (2\pi)^{-d_q/2} e^{\frac{||\mathbf{x}+\mathbf{y}||^2}{2}} \int e^{-\frac{||\omega - \mathbf{x}+\mathbf{y}||^2}{2}} d\omega \\
&= (2\pi)^{-d_q/2} \int e^{-\frac{||\omega||^2 + ||\mathbf{x}+\mathbf{y}||^2 - ||\mathbf{x}+\mathbf{y}||^2 - 2\omega^T \mathbf{x} - 2\omega^T \mathbf{y}}{2}} d\omega \\
&= (2\pi)^{-d_q/2} \int e^{-\frac{||\omega||^2}{2}} e^{\omega^T \mathbf{x}} e^{\omega^T \mathbf{y}} d\omega \\
&= \mathbb{E}_{\omega_i \sim \mathcal{N}(0,\mathbf{I})} (e^{\omega^T \mathbf{x}} e^{\omega^T \mathbf{y}})
\end{aligned} \tag{37}$$

The terms $e^{-||\mathbf{x}||^2}$ and $e^{-||\mathbf{y}||^2}$ in the last line of Eq.36 are independent of $\omega$ and can thus be pushed into the expectation. Finally, we approximate the expectation by sampling in order to get the required result.

**Random Kitchen Sinks:** For the second kernel, we start with Eq. 7.4.6 in Abramowitz & Stegun (1972) and extend it to vectors. We have,

$$\int_{\mathbb{R}^m} e^{-||t||^2} \cos(2\mathbf{t}^T \mathbf{x}) d\mathbf{t}$$

$$= \int_{\mathbb{R}^m} e^{-(t_0^2 + \sum_{i=1}^{m-1} t_i^2)} \cos\left(2x_0 t_0 + 2\sum_{i=1}^{m-1} t_i x_i\right) dt_0 dt_1 \dots dt_{m-1}$$

$$= \int_{\mathbb{R}^m} e^{-(t_0^2 + \sum_{i=1}^{m-1} t_i^2)} \cos(2x_0 t_0) \cos\left(2\sum_{i=1}^{m-1} t_i x_i\right) dt_0 dt_1 \dots dt_{m-1}$$

$$- \int_{\mathbb{R}^m} e^{-(t_0^2 + \sum_{i=1}^{m-1} t_i^2)} \sin(2x_0 t_0) \sin\left(2\sum_{i=1}^{m-1} t_i x_i\right) dt_0 dt_1 \dots dt_{m-1}$$

The second integral involving $\sin$ is odd and therefore evaluates to 0. That leaves us with:

$$= \int_{\mathbb{R}^m} e^{-(t_0^2 + \sum_{i=1}^{m-1} t_i^2)} \cos(2x_0 t_0) \cos\left(2\sum_{i=1}^{m-1} t_i x_i\right) dt_0 dt_1 \dots dt_{m-1}$$

$$= \int_{\mathbb{R}^{m-1}} \left(\int_{-\infty}^{\infty} e^{-t_0^2} \cos(2x_0 t_0) dt_0\right) e^{-\sum_{i=1}^{m-1} t_i^2} \cos\left(2\sum_{i=1}^{m-1} t_i x_i\right) dt_1 \dots dt_{m-1}$$

$$= \frac{1}{2}\sqrt{\pi} e^{-x_0^2} \int_{\mathbb{R}^{m-1}} e^{-\sum_{i=1}^{m-1} t_i^2} \cos\left(2\sum_{i=1}^{m-1} t_i x_i\right) dt_1 \dots dt_{m-1}$$

This process can now be repeated for every dimension of $t$ and $x$ to finally give:

$$\int_{\mathbb{R}^m} e^{-||\mathbf{t}||^2} \cos(2\mathbf{t}^T \mathbf{x}) d\mathbf{t} = \frac{\pi^{m/2}}{2^m} e^{-||\mathbf{x}||^2} \tag{38}$$

Using Eq. 38, we can now get

$$\Phi_{RKS}(\mathbf{X})\Phi_{RKS}(\mathbf{Y}) = \frac{2^m}{k} \sum_{i=0}^{k-1} \left(\cos(\omega_i^T \mathbf{x})\cos(\omega_i^T \mathbf{y}) + \sin(\omega_i^T \mathbf{x})\sin(\omega_i^T \mathbf{y})\right)$$

$$= \frac{2^m}{k} \sum_{i=0}^{k-1} \cos\left(\omega_i(\mathbf{x} - \mathbf{y})\right)$$

$$\approx 2^m \mathop{\mathbb{E}}_{\omega_i \sim \mathcal{N}(\mathbf{0}_m, \mathbb{I})} \cos\left(\omega(\mathbf{x} - \mathbf{y})\right) \tag{39}$$

$$= 2^m \int_{\mathbb{R}^m} \frac{1}{(2\pi)^{m/2}} e^{-\frac{||\omega||^2}{2}} \cos(2\omega^T \frac{\mathbf{x} - \mathbf{y}}{2}) d\omega$$

$$= \frac{2^m}{\pi^{m/2}} \int_{\mathbb{R}^m} e^{-||\frac{\omega}{\sqrt{2}}||^2} \cos(2\frac{\omega}{\sqrt{2}}^T \frac{\mathbf{x} - \mathbf{y}}{\sqrt{2}}) d\frac{\omega}{\sqrt{2}}$$

$$= e^{-\frac{||\mathbf{x} - \mathbf{y}||^2}{2}}$$

In either case, the error stems from the approximation of the expectation by sampling, which can be made arbitrarily small by increasing $k$

### A.7 Lemma S.3

#### A.7.1 Statement

Given,

$$
\begin{aligned}
\mathbf{z}_i^1 = [0,\ldots,0, & c^{(i+1)}, \\
& [\![q^{(i+1)}]\!], [\![v^{(i)}]\!], m^{(i)}, m^{(i-1)}, 0, 0, \\
& \beta^{(i+1)}, \mathbf{0_s}, 0, \\
& 1, (i+1), \frac{1}{(i+1)}, \frac{1}{(i+1)^2}]
\end{aligned}
\tag{40}
$$

we need a construction that gives

$$
\begin{aligned}
Att(W_q^2 \mathbf{z}_i^2, W_k^2 \mathbf{Z}^2, W_v^2 \mathbf{Z}^2) = [0,\ldots,0, \\
0,\ldots,0, \\
0, [\![v^{(\ell(i+1))}]\!], (\ell(i+1)), \\
0, 0, 0, 0]
\end{aligned}
\tag{41}
$$

#### A.7.2 Proof

We set the weight matrices to get $\mathbf{q}_j = W_q^2 \mathbf{z}_j^2 = [0,\ldots,0, c^{(j+1)}, 0, 0]$, $\mathbf{k}_j = W_k^2 \mathbf{z}_j^2 = [0,\ldots,0, c^{(j)} = c^{(j+1)} - m^{(i)}, 0, \frac{1}{(j+1)}]$ and $\mathbf{v}_j = W_v^2 \mathbf{z}_j^2 = [0,\ldots,0, [\![v^{(j)}]\!], j, 0, 0, 0, 0]$. All these are partial permutations and therefore can be done using appropriate binary matrices.

Note that the required output is exactly the value at $j = \ell(i+1)$, so it is sufficient to show that the $score(\mathbf{q}_i, \mathbf{k}_j)$ is maximised in $j$ for $j = \ell(i+1)$, i.e.

$$
j = \begin{cases} \max\{j' | c^{(j')} = c^{(i+1)}\}, & if \ \exists j' \ s.t. \ c^{(j')} = c^{(i+1)} \\ i, & \text{otherwise} \end{cases}
\tag{42}
$$

Now we have $score(\mathbf{q}_i, \mathbf{k}_j) = -(c^{(i+1)} - c^{(j)})^2 - \frac{1}{(j+1)^2}$. For all $j$ such that $c^{(i+1)} \neq c^{(j)}$), the $score$ is almost $-1$ since $c$ is an integer. If there $\exists j' \ s.t. \ c^{(j')} = c^{(i+1)}$ then the corresponding $score$ is greater that $-1$, and the maxima is achieved at the highest such value of $j$. If such $j$ does not exist however, then $\forall j < i, \ score(\mathbf{q}_i, \mathbf{k}_j) < -1 - \frac{1}{(i+1)^2}$ and therefore, the maxima is achieved at $j = i$.

## B Mean Square Error of Linear Approximations

In this section, we calculate the Variance/Mean Square Error(MSE) in the Linear approximation of the Gaussian Kernel. Our proof is based on the similar proof in Choromanski et al. (2021b).

### B.1 Random Kitchen Sinks

#### B.1.1 Statement

For a GMM-RKS estimator with $m$ samples for a Normal distribution with mean vector $\mu$ and Covariance Matrix $\Sigma = S^T S$, the variance of the estimate around its mean is given by:

$$
MSE(\phi_{\text{GMM−RKS}}(q)^T \phi_{\text{GMM−RKS}}(k)) = \frac{2}{m} \cos^2(\mu^T(k-q))(1 - \exp(-||S^T(k-q)||^2))^2
\tag{43}
$$

### B.1.2 Proof

$$MSE(\phi_{\text{GMM}-\text{RKS}}(q)^T \phi_{\text{GMM}-\text{RKS}}(k))$$

$$= \frac{1}{m^2} Var_{\omega_i \sim \mathcal{N}(\mu,\Sigma), \chi_i \sim \mathcal{N}(-\mu,\Sigma)} (\sum_{i=1}^{m} (\cos(\omega_i^T q) \cos(\omega_i^T k) + \sin(\omega_i^T q) \sin(\omega_i^T k)$$

$$+ \cos(\chi_i^T q) \cos(\chi_i^T k) + \sin(\chi_i^T q) \sin(\chi_i^T k)))$$

$$= \frac{1}{m^2} Var_{\omega_i \sim \mathcal{N}(\mu,\Sigma), \chi_i \sim \mathcal{N}(-\mu,\Sigma)} (\sum_{i=1}^{m} (\cos(\omega_i^T (q-k)) + \cos(\chi_i^T (q-k))))$$

$$= \frac{1}{m^2} Var_{\eta_i \sim \mathcal{N}(0,\mathbb{I})} (\sum_{i=1}^{m} (\cos((\eta_i^T S^T + \mu^T)(q-k)) + \cos((\eta_i^T S^T - \mu^T)(q-k)))) \quad (44)$$

$$= \frac{4}{m^2} Var_{\eta_i \sim \mathcal{N}(0,\mathbb{I})} (\sum_{i=1}^{m} \cos(\eta_i^T S^T (q-k)) \cos(\mu^T (q-k)))$$

$$= \frac{4}{m^2} cos^2(\mu^T (q-k)) Var_{\eta_i \sim \mathcal{N}(0,\mathbb{I})} (\sum_{i=1}^{m} \cos(\eta_i^T S^T (q-k)))$$

$$= \frac{4}{m} cos^2(\mu^T (q-k)) Var_{\eta \sim \mathcal{N}(0,\mathbb{I})} \cos(\eta^T (S^T (q-k)))$$

$$= \frac{2}{m} cos^2(\mu^T (q-k))(1 - \exp(-||S^T (q-k)||^2)^2)$$

Note that there is a slight abuse of notation in that $\omega$ and $\chi$ are not independently sampled, but are transforms of the same sample, making the change of variable valid. Since all $\eta$ are iid, we can pull the summation out of the variance. Thereafter we apply Lemma 1 from Yu et al. (2016a) to calculate the final variance.

### B.2 Positive Random Features

For a GMM-PRF estimator with $m$ samples for a Normal distribution with mean vector $\mu$ and Covariance Matrix $\Sigma = S^T S$, the variance of the estimate around its mean is given by:

$$MSE(\phi_{\text{GMM}-\text{PRF}}(q)^T \phi_{\text{GMM}-\text{PRF}}(k))$$
$$= \frac{1}{m} \exp(-2(||q||^2 + ||k||^2 - \mu^T (q+k)))(\exp(2||S^T (q+k)||) - \exp(||S^T (q+k)||)) \quad (45)$$

### B.2.1 Proof

$$MSE(\phi_{\text{GMM}-\text{PRF}}(q)^T \phi_{\text{GMM}-\text{PRF}}(k))$$

$$= \frac{1}{m^2} Var_{\omega_i \sim \mathcal{N}(\mu,\Sigma)} (\sum_{i=1}^{m} (\exp(\omega_i^T (q+k) - ||q||^2 - ||k||^2)))$$

$$= \frac{1}{m} \exp(-2(||q||^2 + ||k||^2)) Var_{\omega \sim \mathcal{N}(\mu,\Sigma)} \exp(\omega^T (q+k))$$

$$= \frac{1}{m} \exp(-2(||q||^2 + ||k||^2)) Var_{\eta \sim \mathcal{N}(0,\mathbb{I})} (\exp(\eta^T S^T (q+k) + \mu^T (q+k))) \quad (46)$$

$$= \frac{1}{m} \exp(-2(||q||^2 + ||k||^2 - \mu^T (q+k))).$$
$$\quad (\mathbb{E}_{\eta \sim \mathcal{N}(0,\mathbb{I})}(\exp(2\eta^T S^T (q+k))) - (\mathbb{E}^2_{\eta \sim \mathcal{N}(0,\mathbb{I})}(\exp(\eta^T S^T (q+k)))))$$

$$= \frac{1}{m} \exp(-2(||q||^2 + ||k||^2 - \mu^T (q+k)))(\exp(2||S^T (q+k)||) - \exp(||S^T (q+k)||))$$

Where the last step follows from Eq. 16 in Choromanski et al. (2021b) which in turn follows from the fact that GMM-PRF is an unbiased estimator for gaussians.

| Model | Image |
|---|---|
| Random Predictor | 10.00 |
| Baseline Models | |
| Softmax Trans.(Vaswani et al.) | 42.44 |
| Synthesizer(Tay et al.) | 41.61 |
| Sinkhorn(Tay et al.) | 41.23 |
| Sparse Trans.(Child et al.) | **44.24** |
| Reformer(Kitaev et al.) | 38.07 |
| Local Attention (Parmar et al.) | 41.46 |
| Longformer(Beltagy et al.) | 42.22 |
| Linformer(Wang et al.) | 38.56 |
| Big Bird(Zaheer et al.) | 40.83 |
| LinearElu(Katharopoulos et al.) | 42.34 |
| Performer(Choromanski et al.) | 42.77 |
| Kernelized Transformers | |
| GMM-RKS (Eq. 5) | 42.33 |
| FASTFOOD-RKS (Eq. 6) | 36.74 |
| GENERATIVE-RKS (Eq. 8) | 39.84 |
| GMM-PRF (Eqs. 9, 10 ) | 39.94 |
| FASTFOOD-PRF (Eqs. 9, 12 ) | 38.31 |
| GENERATIVE-PRF (Eqs. 9, 11 ) | 40.01 |

Table 5: Experimental results on the *Image* dataset with 1024 tokens from *LRA* benchmark.

# C   Experimental Details

## C.1   Source Code

We implemented KL-TRANSFORMERs in Python 3 and PyTorch (Paszke et al., 2019) and plan to open-source the code for reproducing all experiments upon acceptance.

## C.2   LRA image dataset results

For completeness we also give results on *Image* datasets from LRA task where a $N \times N$ image is flattened into a sequence of $N^2$ pixels which is then provided as input to the model. The gray-scaled CIFAR10 image classification dataset (Krizhevsky, 2009) is used, resulting in a sequence length of $1024$.

## C.3   Hyperparameters for LRA Tasks

**Further Notes:**

- To benchmark memory in Figure 3, we used a batch size of 32 for *Text* and a batch size of 2 for *Retrieval*.

- For GMM-RKS and GMM-PRF the number of components $C$ in the mixture was set to $C = 2$.

| Parameter | ListOps | Text | Retrieval | Image |
|---|---|---|---|---|
| Batch Size | 32 | 32 | 32 | 256 |
| Learning Rate | $5 \times 10^{-3}$ | $5 \times 10^{-2}$ | $5 \times 10^{-2}$ | $5 \times 10^{-4}$ |
| Training Steps/Epochs | 10K/NA | 20K/NA | 5K/NA | NA/200 |
| Optimizer | Adam with Weight Decay ($\beta_1 = 0.9, \beta_2 = 0.98$) | | | |
| Weight Decay | 0.1 | 0.1 | 0.1 | 0.0 |
| Warmup Steps | 1000 | 8000 | 8000 | 175 |
| Scheduler | Sqrt Decay | Sqrt Decay | Sqrt Decay | Cosine Decay |
| Loss | Cross Entropy | | | |
| Sequence Length | 2000 | 4000 | 4000 | 1024 |
| Num. Layers | 6 | 4 | 4 | 1 |
| Num. Heads | 8 | 4 | 4 | 8 |
| Embedding Dim. | 512 | 256 | 128 | 128 |
| Key/Query/Value Dim. | 64 | 64 | 32 | 8 |
| Feedforward Dim. | 2048 | 1024 | 512 | 128 |
| Dropout Rate | 0.1 | 0.1 | 0.1 | 0.3 |
| Activation Function | Gelu | Gelu | Gelu | Gelu |
| Positional Encoding | Sinusoidal | Sinusoidal | Sinusoidal | Learnable |
| Pooling Mode | CLS | CLS | CLS | CLS |

Table 6: Hyperparameters for *LRA* tasks.

| Model | ListOps | Text | Retrieval | Image |
|---|---|---|---|---|
| GMM-RKS | 256 | 128 | 64 | 128 |
| FASTFOOD-RKS | 64 | 64 | 32 | 8 |
| GENERATIVE-RKS | 256 | 256 | 128 | 128 |
| GMM-PRF | 256 | 256 | 128 | 128 |
| FASTFOOD-PRF | 64 | 64 | 32 | 8 |
| GENERATIVE-PRF | 128 | 128 | 64 | 8 |

Table 7: Number of random samples $M$ used within each KL-TRANSFORMER.

## C.4 Hyperparameters for GLUE Tasks

| Parameter | Value(s) |
|---|---|
| Pre-Training Batch Size | 64 |
| Batch Size | 64 |
| Pre-Training Learning Rate ($\eta_{\text{pre}}$) | $5 \times 10^{-4}$ |
| Pre-Training Learning Rate at Step $i$ | $\min(\frac{i}{10000}, \frac{I_{\text{pre}}-i}{I_{\text{pre}}-10000}) * \eta_{\text{pre}}$ |
| Training Learning Rate ($\eta_{\text{train}}$) | $\{2 \times 10^{-3}, 1 \times 10^{-4}, 5 \times 10^{-4}, 2 \times 10^{-5}, 5 \times 10^{-6}\}$ |
| Training Learning Rate at Step $i$ ($\eta_{\text{train}}$) | $\min(\frac{10i}{I_{\text{tune}}}, \frac{I_{\text{tune}}-i}{0.9*I_{\text{tune}}}) * \eta_{\text{train}}$ |
| Pre-Training Epochs | 5 |
| Training Epochs | 10 |
| Optimizer | Adam with Weight Decay ($\beta_1 = 0.9, \beta_2 = 0.999$) |
| Weight Decay | 0.01 |
| Loss | Cross Entropy |
| Sequence Length | 512 |
| Num. Layers | 3 |
| Num. Heads | 10 |
| Embedding Dimension | 300 |
| Key/Query/Value Dimension | 64 |
| Transformer Feedforward Dimension | 512 |
| Classifier Feedforward Dimension | 128 |
| Dropout Rate | 0.1 |
| Transformer Activation Function | Gelu |
| Classifier Activation Function | Tanh |
| Positional Encoding | Sinusoidal |
| Pooling Mode | CLS |
| Num. of Samples from Distribution | $\{64, 128\}$ |

Table 8: Hyperparameters for *GLUE* tasks. Where multiple parameters were tried, they are listed in curly brackets. $I_{\text{pre}}$ denotes the total number of pre-training steps, whereas $I_{\text{tune}}$ denotes the total number of fine-tuning steps on each *GLUE* task.

Our model has significantly fewer number of parameters as compared to Devlin et al. (2019) and therefore we perform poorer on than them on all datasets. They use 24 layers with 16 heads each. If reported in the same order as the columns of Table 3, their numbers would look like: 97.5, 89.3/85.4, 72.1/89.3, 87.2/86.4, 92.7, 65.1, 70.1.

While we had to limit model sizes due to resource limitations, this handicaps all models equally, and therefore should not prevent comparison across various models reported in our paper.

## C.5 Further Results on Efficiency Benchmarks

Figure 7 shows how much memory each model uses. Figure 8 plots this memory usage against the model performance

## C.6 Correlation of Variance Metrics

Figures 9 and 10 show the correlation of Average Gain by Voting with the other twp variance metrics for the text and retrieval tasks respectively

| Layer | Head | Minimum | Maximum | Mean |
|-------|------|---------|---------|------|
|       | 1    | 7.601e-7 | 4.584e-1 | 3.531e-2 |
| 1     | 2    | 2.380e-8 | 4.863e-1 | 3.576e-2 |
|       | 3    | 1.400e-5 | 4.469e-1 | 3.319e-2 |
|       | 4    | 4.214e-6 | 4.197e-1 | 3.208e-2 |
|       | 1    | 4.242e-6 | 2.790e-1 | 3.070e-2 |
| 2     | 2    | 2.971e-6 | 3.238e-1 | 3.161e-2 |
|       | 3    | 5.081e-6 | 2.842e-1 | 3.078e-2 |
|       | 4    | 5.212e-7 | 3.006e-1 | 3.086e-2 |
|       | 1    | 2.297e-6 | 2.219e-1 | 2.872e-2 |
| 3     | 2    | 7.094e-6 | 2.036e-1 | 2.594e-2 |
|       | 3    | 1.074e-7 | 1.953e-1 | 2.772e-2 |
|       | 4    | 6.330e-9 | 1.836e-1 | 2.442e-2 |

Table 9: Distribution of Eigenvalues of covariances of GMM-RKS model for the *Text* task

| Layer | Head | Minimum | Maximum | Mean |
|-------|------|---------|---------|------|
|       | 1    | 1.363e-5 | 9.660e-2 | 2.440e-2 |
| 1     | 2    | 6.659e-5 | 7.900e-2 | 2.200e-2 |
|       | 3    | 7.241e-6 | 9.050e-2 | 2.810e-2 |
|       | 4    | 2.760e-5 | 7.790e-2 | 2.630e-2 |
|       | 1    | 2.068e-5 | 7.520e-2 | 2.430e-2 |
| 2     | 2    | 2.386e-5 | 7.580e-2 | 2.400e-2 |
|       | 3    | 7.256e-6 | 7.400e-2 | 2.370e-2 |
|       | 4    | 1.911e-4 | 8.080e-2 | 2.680e-2 |
|       | 1    | 2.871e-5 | 6.710e-2 | 2.570e-2 |
| 3     | 2    | 6.117e-6 | 7.370e-2 | 2.190e-2 |
|       | 3    | 7.830e-6 | 6.600e-2 | 2.010e-2 |
|       | 4    | 2.186e-5 | 7.620e-2 | 2.810e-2 |

Table 10: Distribution of Eigenvalues of covariances of GMM-PRF model for the *Text* task

| Layer | Head | Minimum | Maximum | Mean |
|-------|------|---------|---------|------|
|       | 1    | 4.011e-05 | 4.994e-01 | 5.998e-02 |
| 1     | 2    | 4.250e-05 | 2.294e-01 | 5.390e-02 |
|       | 3    | 1.918e-04 | 3.427e-01 | 5.842e-02 |
|       | 4    | 5.615e-06 | 3.943e-01 | 5.866e-02 |
|       | 1    | 3.232e-06 | 2.605e-01 | 5.469e-02 |
| 2     | 2    | 2.746e-06 | 2.581e-01 | 5.333e-02 |
|       | 3    | 6.364e-06 | 2.329e-01 | 5.091e-02 |
|       | 4    | 2.071e-05 | 1.743e-01 | 5.020e-02 |
|       | 1    | 4.358e-05 | 1.919e-01 | 5.157e-02 |
| 3     | 2    | 1.143e-04 | 1.738e-01 | 4.819e-02 |
|       | 3    | 2.546e-05 | 1.735e-01 | 4.705e-02 |
|       | 4    | 9.697e-07 | 2.075e-01 | 4.982e-02 |

Table 11: Distribution of Eigenvalues of covariances of GMM-RKS model for the *Retrieval* task

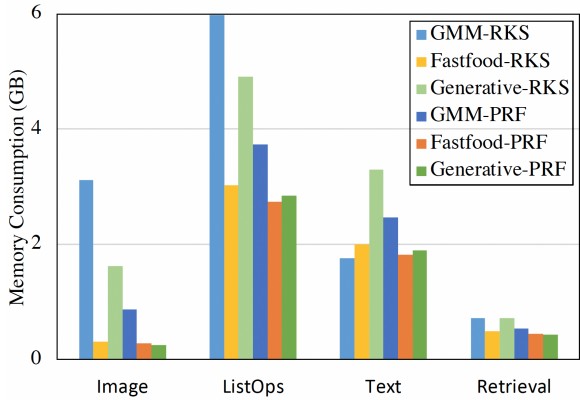

Figure 7: Peak memory used by KL-TRANSFORMERs across different datasets.

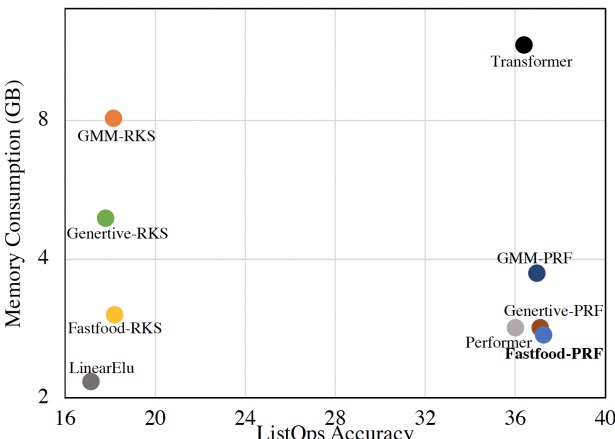

Figure 8: We demonstrate the peak memory consumption (y-axis) and performance (x-axis) of the various Kernelized Transformer architectures on the ListOps dataset from LRA. Memory usage refers to per device memory usage across each GPU.

### C.7 Eigenvalues of Trained Models

## D Ablation Studies

### D.1 FastFood Attention

In the main paper, we use FastFood-SGB, which has all the diagonal matrices learnable. However, $B$ and $G$ matrices have a very special structure (their elements being drawn from $Bernoulli_{\{-1,1\}}(0.5)$ and $\mathcal{N}(0,1)$ respectively), which is lost if we make them learnable. Therefore, it makes sense to have *FastFood-S*, which only has $S$ learnable. Finally, we can also have everything fixed, giving us the basic *FastFood* version. The results of these two versions, along with the original FastFood-SGB kernel on the *GLUE* benchmaark are summarised in Table 13. As one can see, FastFood-SGB is either the best or close to it except for WNLI and CoLA, therefore we choose to use this version for our main analysis.

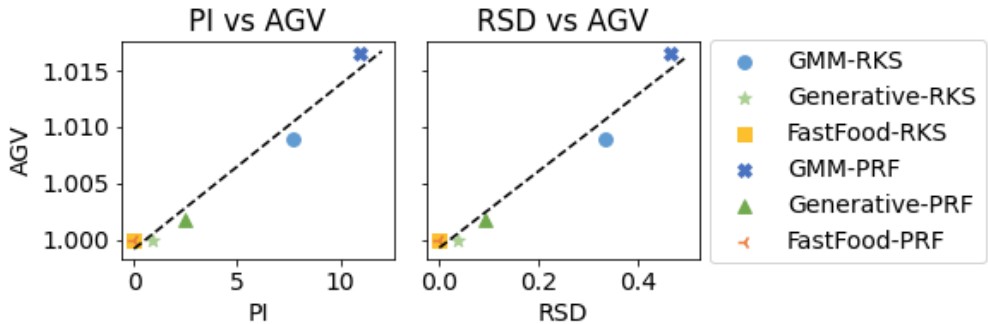

Figure 9: Correlation between AGV and the other two variance metrics on the **Text** task.

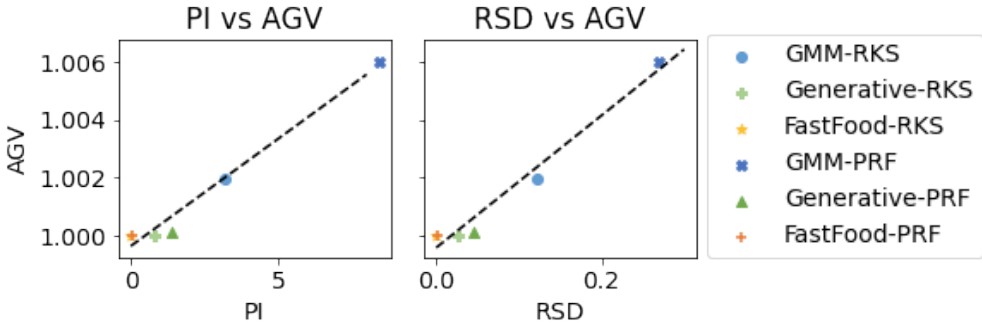

Figure 10: Correlation between AGV and the other two variance metrics on the **Retrieval** task.

.

# E   Sparsity Synthetic Experiment

## E.1   Task Description

Given a sequence of ordered pairs $(v_i, a_i)$, where $v_i \in \{-1, 1\}$ and $a_i \in \{0, 1\}$, the task is to output $\sum_{i=0}^{L} v_i a_i$. Here, $v_i$ can be seen as the value of a given position, while $a_i$ indicates whether or not we need to attend to that position. The dataset is generated by (pseudo-randomly flipping a coin independently for $a_i$ and $v_i$. The bias in the flip for $a_i$ defines the sparsity of the dataset. The flip for $v_i$ is appropriately biased to ensure that no prefix has an absolute sum of more that 4. The final prediction is outputted as a 9-way classification over integer values between $-4$ and 4 (both

| Layer | Head | Minimum | Maximum | Mean |
|-------|------|---------|---------|------|
| 1 | 1 | 8.225e-05 | 2.114e-01 | 4.818e-02 |
|   | 2 | 5.386e-05 | 1.525e-01 | 4.066e-02 |
|   | 3 | 9.659e-05 | 1.707e-01 | 5.638e-02 |
|   | 4 | 1.498e-07 | 1.467e-01 | 4.166e-02 |
| 2 | 1 | 1.294e-06 | 9.798e-02 | 3.024e-02 |
|   | 2 | 9.306e-05 | 1.289e-01 | 4.334e-02 |
|   | 3 | 4.300e-06 | 1.865e-01 | 4.846e-02 |
|   | 4 | 2.418e-05 | 1.047e-01 | 4.458e-02 |
| 3 | 1 | 2.466e-08 | 1.133e-01 | 3.987e-02 |
|   | 2 | 2.291e-04 | 1.429e-01 | 5.099e-02 |
|   | 3 | 3.997e-05 | 1.134e-01 | 3.380e-02 |
|   | 4 | 6.653e-05 | 1.129e-01 | 2.827e-02 |

Table 12: Distribution of Eigenvalues of covariances of GMM-PRF model for the *Retrieval* task

| Dataset | SST2 (acc) | MRPC (acc) | MRPC (f1) | QQP (acc) | QQP (f1) | MNLI (mat) | MNLI (mis) | QNLI (acc) | WNLI (acc) | RTE (acc) | CoLA (MCorr) |
|---|---|---|---|---|---|---|---|---|---|---|---|
| FastFood | 0.814 | 0.713 | 0.820 | 0.811 | 0.738 | 0.571 | 0.568 | 0.629 | 0.634 | 0.563 | 0.152 |
| FastFood-S | 0.807 | 0.706 | 0.822 | 0.810 | 0.741 | 0.571 | 0.571 | 0.642 | 0.606 | 0.570 | 0.101 |
| FastFood SGB | 0.828 | 0.707 | 0.820 | 0.810 | 0.739 | 0.569 | 0.572 | 0.638 | 0.592 | 0.563 | 0.129 |

Table 13: Ablation studies using FastFood variants on the *GLUE* benchmark.

| Dataset Sparsity | Checkpoint Acc.(%) | GMM-RKS | | GMM-PRF | | GMM-RKS/GMM-PRF | |
|---|---|---|---|---|---|---|---|
| | | Std. Dev. | Abs. Mean | Std. Dev. | Abs. Mean | Std. Dev. | Abs. Mean |
| 0.1 | 20 | 0.083064 | 0.144132 | 0.079822 | 0.139948 | 0.960974 | 0.97097 |
| | 40 | 0.094528 | 0.16609 | 0.11077 | 0.193444 | 1.171828 | 1.164689 |
| | 60 | 0.099791 | 0.176456 | 0.116044 | 0.201462 | 1.162868 | 1.141712 |
| | 80 | 0.131252 | 0.228456 | 0.154802 | 0.275594 | 1.179433 | 1.206332 |
| 0.5 | 20 | 0.069646 | 0.121306 | 0.434024 | 0.53019 | 6.231835 | 4.370671 |
| | 40 | 0.11552 | 0.201468 | 6.432806 | 12.17245 | 55.68558 | 60.41891 |
| | 60 | 0.142631 | 0.249917 | 3.752173 | 5.886064 | 26.30681 | 23.55206 |
| | 80 | 0.166899 | 0.295308 | 22.46943 | 42.14157 | 134.6289 | 142.7036 |
| 0.9 | 20 | 0.059153 | 0.102415 | 0.065438 | 0.115866 | 1.106237 | 1.131343 |
| | 40 | 0.108255 | 0.186704 | 0.123171 | 0.211465 | 1.137784 | 1.132622 |
| | 60 | 0.119243 | 0.206855 | 0.126247 | 0.217067 | 1.058737 | 1.049365 |
| | 80 | 0.136816 | 0.240525 | 0.168633 | 0.295976 | 1.232546 | 1.230542 |

Table 14: Means and Variances of gradients received at the classifier layer for the synthetic experiment

inclusive). The bias against higher absolute values causes $-4$ and $4$ classes to appear less often. This is balanced out by overgeneration and sampling.

We generate 3 datasets corresponding to sparsities $0.1$, $0.5$ and $0.9$. Each dataset has $200K$ instances, of sequence length 200. Of these, we use $80\%$ as the training set and the rest for validation.

## E.2 Model Description

We use a 3 layer transformer with $d_{model} = d_{feedforward} = 64$ and $d_{query} == d_{value} = 16$. The input is encoded as a 3-d many-hot vector $(v_i = 1, v_i = -1, a_i)$. this is then passed through an embedding layer and added to learnable position embeddings and passed through the transformer. The final embedding of the $0^{th}$ position is then passed through a hidden layer with 64 units and then passed on to the final layer for a 9-way softmax. For both linear attention models, we use 64 samples.

All models are trained using AdamW optimizer with $\beta_2 = 0.98$, $\epsilon = 10^{-9}$, weight decay=0.1, Learning Rate=$5 \times 10^{-6}$ and all other parameters set to default. We use SGD with a batch size of 400 and cross entropy loss.

## E.3 Gradients

We create checkpoints for the GMM-RKS and GMM-PRF models when they first achieve validation accuracies of 20%, 40%, 60% and 80%. For each of these checkpoints, we pass the first 50 validation datapoints and record the gradients on the classifier layer. This process is repeated 50 times. Thereafter, we calculate the mean and standard deviation of gradients to each neuron. To avoid cancellation of opposite signs, the mean is calculated over absolute values. The final reported numbers are averages over the 64 neurons. The results are shown in Table 14

