# OpenReview forum: "Learning the Transformer Kernel"
_TMLR — Accepted by TMLR_

### Review · Reviewer_x9TU · 2022-05-13

**Summary Of Contributions:**

The paper builds on the idea of linear transformers, which reformulated the standard attention mechanism by rewriting it in terms of kernel functions. In this paper, the kernel is approximated with random Fourier features (RFFs), and the underlying spectral distribution is learnt with different mechanisms. A few theoretical claims are made about the method (notably, that it keeps Turing completeness of the original mechanism). Experiments show that the method performs better (in terms of accuracy and/or speed) compared to some baselines.

**Broader Impact Concerns:**

I do not see significant ethical concerns here.

**Requested Changes:**

The only part of the paper which is not clear to me is the use of generative models (Fig. 2). If the generator is trained together with the rest of the network, sampling should be made differentiable, but this not mentioned. If it not trained together (somewhere in the text it is mentioned that samples are generated every fixed number of iterations), then, how is it trained?

If I understand correctly, a different kernel is learnt for every head in the network. It would be interesting to check whether it is possible to learn a shared distribution for the heads, or even for different layers.

**Strengths And Weaknesses:**

Strenghts:
- The idea is sound and the paper is easy to read, except for one contribution (see below).
- Experiments are comprehensive, except for the efficiency evaluation (again, see below).
- Theoretical claims are clear, but I have not checked the full derivations in the appendix.

Weaknesses:
- The novelty can be considered minor, since the idea of using RFFs inside transformers is known, and the methods they use to learn the kernel approximation are all leveraged from the corresponding literature, with only one exception.
- The paper seems to be stuck at mid-2021 in terms of citations, comparisons, related works, etc. I am assuming this is a prior submission that was not updated.
- The use of deep generative techniques is not clear (see below on requested changes).
- I do not understand why the efficiency comparison is made by considering a different set of methods compared to the accuracy claims.

Overall, I would vote for accepting the paper if these points are addressed.

---

> ### Author Response · Authors · 2022-06-05
> **Reply From Authors**
>
> We would like to thank the reviewer for their comments. We try to address as many of them as possible below
> * There are indeed some missing citations, due to works that have been published between now and when we did the literature survey. We have conducted a new literature survey, and these papers have been added in our updated draft.
> * The efficiency comparisons shown in figure 3 in the main paper and figure 8 in the supplementary include all six of our proposed methods and 3 representative baselines (Transformers, LinearElu and Performers). We left out other baselines from this figure to avoid clutter. Figure 1 shows only our best performing model and includes comparisons to all baselines shown in Table 2. However, should the reviewer wish, we are happy to replace figure 3 with bigger figures which contain all the baselines in our next version.
> * We thank the reviewer for pointing this out, upon rereading, we find that the explanation for the generator model is indeed somewhat lacking. We propose to change the second paragraph of the paper to better explain this, as can be seen in the updated version. We hereby request the reviewer to let us know if the proposed explanation is more clear, and to let us know in case some part needs more clarification

---

> > ### Comment · Reviewer_x9TU · 2022-06-06
> > **Response to revised manuscript**
> >
> > I thank the authors for carefully addressing all my comments. Regarding the final point (which was my main issue), I believe the paper is much clearer now. I have no specific comments on the revised version.

---

### Review · Reviewer_fZ1L · 2022-05-15

**Summary Of Contributions:**

This paper proposes to learn the kernel of the self-attention layer in Transformer architectures. Inspired by kernel methods and random Fourier feature models, the authors approximate the kernel function with explicit feature maps by sampling from a spectral distribution. For computational reasons, the authors further approximate the spectral distribution with either a mixture of Gaussians (with a clever parameterization) or a flexible generator. As a consequence, the resulting kernelized transformer attains a linear complexity in terms of the sequence length.

In theory, the authors proved that the resulting architecture with positional encodings is Turing-complete. In practice, the model with learnt kernels improves the performance of long-context tasks, while staying competitive to standard transformer with softmax kernel in short-context tasks.

**Broader Impact Concerns:**

The main contribution of the paper is on making the transformer architecture more computationally efficient, so it is more on the methodology. I do not expect there to be any direct negative societal impact rom this contribution. In addition, all experiments were done in public datasets. That being said, a better transformer model might be used in many applications (e.g., language model) with toxic outputs. However, this is out of the scope of this work.

**Requested Changes:**

Overall, I like this paper. Albeit the idea of learning the kernel is straightforward, this work makes solid contributions to reducing the complexity of the self-attention module while retaining the performance.

My main request is to remove these flimsy arguments (as mentioned in weaknesses). Many conclusions in the paper are merely scientific guesses or conjectures.

The authors have discussed the variance of the kernelized attention module, but less is done for bias analysis. I think empirical bias analysis should be easy and it is quite interesting to see how the approximation quality or model performance changes as a function of the number of samples. To further reduce the bias, I recommend the authors check out the Russian Roulette Estimator of approximating an infinite sum. See for example [1].

Reference:
1. Efficient optimization of loops and limits with randomized telescoping sums. ICML 2019.


**Strengths And Weaknesses:**

Strengths:
1. The paper is very well-written and easy to follow.
2. This paper is "complete" with clear motivations, efficient methods, detailed analyses, and strong empirical results. Overall, this paper attempts to attack an important problem of quadratic time complexity in the self-attention module. Though there are many other existing works already, this paper makes distinct contributions by introducing learned kernels and efficient implementations.
3. The authors have conducted comprehensive empirical studies with different tasks and situations. Overall, the empirical results seem to be convincing, but I'm not an expert and are not familiar with the datasets used in this paper.

Weaknesses:
1. Mistakes in time and space complexity for FastFood: if I understand correctly, FastFood parameterization reduces the time complexity to logarithmic time, but this is not reflected in Table 1.
2. Discussions of the empirical results: in Section 3.1, the authors present the results on long sequences with improved performance. However, I find that the results are quite different for text and retrieval. While the RKS kernel is better for text, PRF is better for retrieval. I feel some discussions are required. The authors do give an explanation based on the variance analysis, but it is more of a guess. Besides,  both GMM PRF and generative PRF underperform compared to Performer and LinearElu, which is concerning. I think the authors need to dig a bit deeper and give a clear recommendation to other users.
3. Some conclusions are a little flimsy. In terms of the variance analysis, it seems that the PRF kernel has a higher variance, but on the other hand, it has smaller eigenvalues. I don't know how the authors reach the conclusion that it is consistent with Theorem 3. Also, it is unclear about the casual relationship between variance and performance. Often the case, high variance leads to slow convergence in training, but not necessarily bad performance. To make any scientific arguments (or conclusions), the authors need to design control experiments to verify, rather than just loosely connecting them.

---

> ### Author Response · Authors · 2022-06-05
> **Reply From Authors**
>
> We thank the reviewer for pointing out some concerns in our paper. We attempt to address some of them here:
> * FastFood indeed reduces the time and space complexity of calculating \phi(q) and  \phi(k) from O(Md_qL) to O(d_qL\log(M)). However, we still need to calculate the matrix product between \phi(q) and \phi(k)V which takes time in the order O(Md_qL). So although the actual time taken does reduce with a GPU implementation of the hadamard product, the order is still dominated by the matrix product, and remains unchanged.
> * While theorem 3 states that both RKS and PRF variance is positively related with the eigenvalues of S, their rate of change is quite different. To better see this, if we assume both o and p to contain all the eigenvectors of S, we can get ||So||=||Sp||=mean of eigenvalues of S. Let us call this quantity x. Then the last factor in the variance of RKS grows as (1-e^(-x^2))^2 while the last factor in the variance of PRF grows as e^(2x)-e^x. It is easy to see that while the first equation produces an inverted bell curve which plateaus pretty quickly at 1, the second quantity has an exponential growth rate and goes to infinity. Due to this significantly higher growth rate, we could expect PRF to have a higher variance despite somewhat lower eigenvalues. Note that this example is an oversimplification, as it ignores the contributions from the rest of the factors, and also setting o=p forces q to be zero, but the principle holds in general case too.
> * We left out bias analysis from our experiments since both RKS and PRF have been theoretically shown to be unbiased estimators by their respective papers. We do agree that testing how well this would hold in practice is an interesting venture, but due to the limitation of time and space, we have to leave it to future work.
> * In a fully convex objective function, with a suitable learning rate, any unbiased estimator of variance is guaranteed to lead to convergence, irrespective of their variance. However, in more complex objective functions, such as the ones in LRA, high amount of noise in the variance estimate makes it more likely for the model to fail to find a good local optimum, and we suspect that this is the issue with PRF. We also observe that in the simpler case of the synthetic experiment, PRF always converges, although it takes longer than RKS. We do understand that these explanations are mostly educated guesses, but at this time we are not sure how to verify this further. If the reviewer and editor prefer, we are also open to moving this piece of analysis to the appendix.
> * On analyzing the final models, we found that the average difference between the logits for PRF in the text task are comparable to the standard deviation in the model outputs. What this essentially means is that the separator learnt by the PRF model does have a fairly small margin and since its output is noisy, it cannot predict the class with high accuracy. While this part is clear from the empirical analysis, the real question is why the model could not learn a classifier with a larger margin. Again, we, unfortunately, cannot provide a good answer to this question, and have to chart it to being an artefact of the dataset.
> \item We do understand that some of the conclusions in section 4 do not have strong theoretical ground to stand on, but the goal of this section was to give the reader an idea of the observed strengths and weaknesses of the various models, allowing them to in turn make an educated guess about which model to use. We fail to provide concrete orderings between the quality of our models. We hope to improve on our recommendations in further work, but the theoretical uncertainties, which plague all research in transformers, might take some time to resolve.

---

> > ### Comment · Reviewer_fZ1L · 2022-06-16
> > **Response**
> >
> > I thank the authors for addressing my comments. Most of my concerns are well addressed, however I hope the authors could remove those flimsy conjectures in the paper or provide more evidence.

---

### Review · Reviewer_tXbd · 2022-06-01

**Summary Of Contributions:**

The authors present a new transformer. As proposed in Tsai et al (2019), the self-attention mechanism is here defined using a kernel function to evaluate the similarities of keys and queries. The approach approximates the transformer kernel as a dot product of spectral feature maps, similar to Choromanski et al. (2021). The approach's core novelty (given in Section 2.1) is a method to learn the transformer's kernel. To due so, the authors invoke the framework for kernel learning using GMMs in regression and classification tasks by Oliva et al. (2016) and adapt it to transformers. The proposed transformer is shown to achieve higher accuracy on the LRA benchmark than several baseline methods (including Performer), while having on par (or better) computational complexity. Further experiments reveal a tradeoff in accuracy and efficiency and analyze the effectiveness of sparse data. The appendix shows that the method's accuracy can also be (slightly) inferior to some baseline models (e.g., Table 5). A theoretical analysis is presented showing that the proposed method(s) are Turing complete and giving an analytical characterization of the MSE of the approximation. Proofs are shown in the appendix.


**Broader Impact Concerns:**

-

**Requested Changes:**

Minor comments:
- Describe novelty better and delineate from related work. (Your core contribution seems the aspect of learning the kernel. However, the name "We propose the kernelized transformer" indicates as if you were the first to use kernel methods in transformers.)
- PRF is defined twice on page 2
- space missing before parentheses in Table 2

**Strengths And Weaknesses:**


The paper is well written and structured. The problem is clearly motivated and abstracted. Related work is discussed sufficiently. Although the ingredient are not new, the overall approach gives sufficient noveltely and is shown that it can work more accurately than strong baselines. The computational complexity is linear. A sound and non-trivial theoretical analysis is presented. Although the novelty of the approach could be described a bit better and delineated from the related work, all in all I find this a sound and relevant contribution to the field, worth publishing in TMLR.

---

> ### Author Response · Authors · 2022-06-05
> **Reply From Authors**
>
> We thank the reviewer for their encouraging words. We shall remove the double definition of PRF from page 2 and also fix the formatting of Table 2. As for the name Kernelized Transformer, we intended to use it as a name of our method, and not necessarily a proper definition of our contributions, however, we see how it could be misleading. We propose to replace it with KL Transformer, as can be seen in our updated version

---

> > ### Comment · Action_Editors · 2022-06-21
> > **Can reviewer tXbd respond to authors?**
> >
> > Hi Reviewer tXbd, the authors have responded to your review.  It would be highly appreciated if you could reply with an indication of whether their response is satisfactory or whether you have additional concerns that you feel are unaddressed.  Rather than a simple accept/reject decision, it is also possible to ask the authors to adjust claims made in the paper to match the evidence provided.  Thanks for your quick feedback.

---

> > ### Comment · Reviewer_tXbd · 2022-06-27
> > **Feedback on authors' response.**
> >
> > The authors' answer is satisfactory for me.

---

### Comment · Action_Editors · 2022-06-02
**Beginning of the discussion period**

Thank you for your reviews on this paper.  We will now begin the discussion period which lasts two weeks.  The goal of this period is for the authors to respond to the points raised by reviewers in order to give them sufficient information to make an informed decision recommendation on the paper at that point.

---

### Decision · Action_Editors · 2022-06-26

**Recommendation:** Accept with minor revision

**Comment:**

The reviewers were generally appreciative of the article and in favor of acceptance.  Reviewer fZ1L has remaining concerns about whether claims about the variance are supported.  In particular, they disagree with the logical entailment of the claim that the empirical results are consistent w/ Theorem 3 (last sentence of 4.1), and "it is unclear about the casual relationship between variance and performance" (point 3 of "weaknesses" in their original review).  These claims should be modified in the final version.